# Return of ChebNet: Understanding and Improving an Overlooked GNN on Long-Range Tasks

**Ali Hariri**[1,*,†]    **Álvaro Arroyo**[2,*]    **Alessio Gravina**[3,*]    **Moshe Eliasof**[4]

**Carola-Bibiane Schönlieb**[4]    **Davide Bacciu**[3]    **Kamyar Azizzadenesheli**[5]

**Xiaowen Dong**[2]    **Pierre Vandergheynst**[1]

## Abstract

ChebNet, one of the earliest spectral GNNs, has largely been overshadowed by Message Passing Neural Networks (MPNNs), which gained popularity for their simplicity and effectiveness in capturing local graph structure. Despite their success, MPNNs are limited in their ability to capture long-range dependencies between nodes. This has led researchers to adapt MPNNs through *rewiring* or make use of *Graph Transformers*, which compromises the computational efficiency that characterized early spatial message-passing architectures, and typically disregards the graph structure. Almost a decade after its original introduction, we revisit ChebNet to shed light on its ability to model distant node interactions. We find that out-of-box, ChebNet already shows competitive advantages relative to classical MPNNs and GTs on long-range benchmarks, while maintaining good scalability properties for high-order polynomials. However, we uncover that this polynomial expansion leads ChebNet to an unstable regime during training. To address this limitation, we cast ChebNet as a stable and non-dissipative dynamical system, which we coin `Stable-ChebNet`. Our `Stable-ChebNet` model allows for stable information propagation, and has controllable dynamics which do not require the use of eigendecompositions, positional encodings, or graph rewiring. Across several benchmarks, `Stable-ChebNet` achieves near state-of-the-art performance.

## 1 Introduction

Graph Neural Networks (GNNs) [81, 43, 74, 67, 14, 25, 44] have emerged as a prevalent framework for handling data defined on graphs. Graph convolutional networks have their roots in spectral approaches that extend convolutional filters to non-Euclidean domains. The first practical instantiation of a GNN was proposed by [14], which leveraged the eigenbasis of the graph Laplacian to perform spectral filtering, as an attempt to generalize image convolutions to non-euclidean structures. However, this formulation required costly eigen-decompositions at each layer. Defferrard *et al.* [25] addressed this inefficiency by approximating spectral filters with truncated Chebyshev polynomials, giving rise to *ChebNet*, the first tractable and localized spectral GNN. By parameterizing filters as $K$-order polynomials of the Laplacian, ChebNet could aggregate information from $K$-hop neighborhoods without repeated eigendecompositions, thereby enabling scalable spectral convolution on large graphs.

[1]École Polytechnique Fédérale de Lausanne (EPFL)

[2]University of Oxford

[3]University of Pisa

[4]University of Cambridge

[5]NVIDIA Research

[*]Equal contribution

[†]Correspondence to: Ali Hariri (`ali.hariri@epfl.ch`)

39th Conference on Neural Information Processing Systems (NeurIPS 2025).

In 2017, Kipf and Welling distilled ChebNet into a simpler, first-order approximation now known as the *Graph Convolutional Network (GCN)* by (i) restricting the polynomial order to one and (ii) tying filter coefficients across hops [57]. This yielded an architecture that was both lightweight and effective: a GCN with few layers would achieve strong node-classification performance on standard homophilic benchmarks, and its $\mathcal{O}(|E|)$ complexity made it practical for large-scale graphs. GCN's efficiency and strong locality bias quickly made it the default baseline, and subsequent *message-passing neural networks (MPNNs)* adopted a similar paradigm of iterative neighborhood aggregation [41].

Despite their popularity, MPNNs exhibit pronounced shortcomings when made deeper and when capturing long-range dependencies [33]. Repeated neighborhood aggregations tend to cause *representational collapse*, where node features become indistinguishable (often referred to as "oversmoothing") [15, 69], and information from distant nodes is "squashed" through narrow bottlenecks, limiting the ability to model global context [1]. In recent years, researchers have proposed various strategies to overcome these limitations. To mitigate oversmoothing, several models have drawn

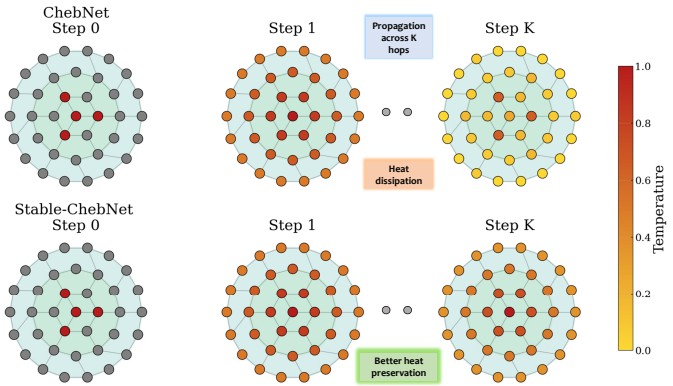

Figure 1: **Top:** Vanilla ChebNet. **Bottom:** Stable-ChebNet. While the original ChebNet's high-order Chebyshev filters induce unstable dynamics resulting in dissipative behavior, our Stable-ChebNet yields bounded propagation through layers.

on principles from physics [28, 12] to preserve feature diversity across layers. At the same time, efforts to capture long-range dependencies have led to graph-rewiring techniques [82, 49, 7] that add or reweight edges to shorten information pathways, as well as the emergence of graph transformers [30], which replace purely local aggregation with global self-attention mechanisms. Although these advances can alleviate depth-related pathologies, they often trade off numerous benefits that made MPNNs appealing: scalability, parameter efficiency, and to only process information along the graph's edges. In this context, ChebNet and other spectral GNNs are usually relegated to a footnote - mentioned only as a predecessor to GCN, which is typically not revisited as a competitive baseline.

In this work, we revisit ChebNet from first principles. We demonstrate that the original Cheb-Net (without any rewiring or attention mechanisms) already delivers state-of-the-art performance or is close on long-range graph tasks while scaling gracefully to large graphs. By deriving and analyzing ChebNet's linearized dynamics, we prove that enlarging its receptive field introduces signal-propagation instabilities. To overcome this, we propose `Stable-ChebNet`, a minimal set of architectural modifications that restore stable propagation for arbitrarily large receptive fields, supported by both theoretical guarantees and empirical validation. To build intuition for our Stable-ChebNet framework, Figure 1 illustrates how classical ChebNet filters (top) can exhibit unbounded dynamics, whereas our antisymmetric, forward-Euler discretization yields smooth, stable propagation (bottom). An intuitive heat-transfer analogy helps explain the difference: if we inject 'heat' at seed nodes, the high-order filters of vanilla ChebNet diffuse and dissipate this heat so that distant nodes cool rapidly. In contrast, the non-dissipative dynamics induced by the antisymmetric, forward-Euler step in Stable-ChebNet preserve energy, keeping temperatures higher at nodes many hops away.

Across a suite of challenging long-range node- and graph-level benchmarks, `Stable-ChebNet` matches or outperforms state-of-the-art message-passing neural networks and graph transformers, while retaining the ChebNet backbone. We hope this work will reignite interest in spectral GNNs as a scalable, theoretically grounded alternative for long-range graph modeling.

**Contributions and Outline**

- In Section 3.1, we empirically demonstrate that vanilla ChebNet can achieve very strong performance on long-range benchmarks without incurring prohibitive computational cost.
- In Section 3.2, we analyze ChebNet's signal-propagation dynamics, providing exact sensitivity analysis, and theoretically and empirically prove the emergence of instability for large filter order.

- In Section 3.3, we introduce `Stable-ChebNet`, which enforces layer-wise stability.
- In Section 4, we empirically validate that `Stable-ChebNet` consistently outperforms MPNNs, rewiring methods, and graph transformers across a number of tasks.

## 2  Background

### 2.1  Background on Spectral Graph Neural Networks

Spectral GNNs extend the notion of convolution to graphs by leveraging the eigen-decomposition of the graph Laplacian. Given an undirected graph $G = (V, E)$ with normalized Laplacian $\mathbf{L} = \mathbf{I} - \mathbf{D}^{-1/2}\mathbf{A}\mathbf{D}^{-1/2}$, any graph signal $\mathbf{X} \in \mathbb{R}^n$ can be filtered in the spectral domain via $\mathbf{Y} = \mathbf{U}g_{\boldsymbol{\theta}}(\boldsymbol{\Lambda})\mathbf{U}^{\top}\mathbf{X}$, where $\mathbf{L} = \mathbf{U}\boldsymbol{\Lambda}\mathbf{U}^{\top}$ diagonalizes the Laplacian, $\boldsymbol{\Lambda} = \mathrm{diag}(\lambda_1, \ldots, \lambda_n)$ its eigenvalues, and $g_{\boldsymbol{\theta}}$ is a learnable spectral response. Early methods directly parameterize $g_{\boldsymbol{\theta}}(\Lambda)$, but require an expensive eigendecomposition of the Laplacian, which can be computationally and memory intensive [14]. ChebNet alleviates the cost of an explicit eigendecomposition by approximating $g_{\boldsymbol{\theta}}$ using the recurrence relation for a $K$-th order Chebyshev polynomial in $\mathbf{L}$ [25]. The latter defines $g_{\boldsymbol{\theta}}$ as $g_{\boldsymbol{\theta}}(\Lambda) \approx \sum_{k=0}^{K} \boldsymbol{\Theta}_k T_k(\tilde{\Lambda})$, where $T_k(\tilde{\Lambda})$ is the $k$-th polynomial of $\tilde{\Lambda}$ with $\tilde{\Lambda} = \frac{2\Lambda}{\lambda_{max}} - \mathbf{I}_n$ . The spectral convolution can then be written without any eigendecomposition as the truncated expansion:

$$\mathbf{Y} = \sum_{k=0}^{K} \boldsymbol{\Theta}_k T_k(\tilde{\mathbf{L}})\, \mathbf{X} \tag{1}$$

where $\tilde{\mathbf{L}} = \frac{2\mathbf{L}}{\lambda_{max}} - \mathbf{I}_n$ enabling efficient, localized filtering in $\mathcal{O}(K|E|)$ time.

### 2.2  MPNNs and their Limitations

Message-Passing Neural Networks (MPNNs) define a general framework in which node features are iteratively updated by exchanging "messages" along edges. At each layer $l$, every node $v$ aggregates information from its neighbors $u \in \mathcal{N}(v)$ and combines it with its own representation. This formulation unifies many graph models, including graph convolutional networks (GCNs) [57] and graph attention networks (GATs) [83]. While this local neighborhood aggregation captures structural information effectively, it has a limited capacity to model long-range interactions within the graph. This is due to the phenomenon of *over-squashing*, an information bottleneck that impedes effective information flow among distant nodes [1, 82, 27]. Numerous techniques have emerged to address this limitation such as graph rewiring [49, 7], Graph Transformers [71, 80, 79] in addition to some enhanced spatial methods that tackle over-squashing through combined local and global information [75, 40], or through non-dissipativity achieved by antisymmetric weight parameterization [45, 46] or port-Hamiltonian systems [54].

However, some of the aforementioned methods suffer from substantial overhead due to denser graph shift operators or the use of all-pairs interactions. Specifically, [79] and other graph transformers increase computational complexity through dense attention-maps; [40] relies on costly full eigendecomposition operations; and graph rewiring techniques heavily pre-process the graph topology, incurring $\mathcal{O}(n^3)$ time in the case of [49]. For a detailed discussion on the relevant literature, we point the reader to the Appendix A.

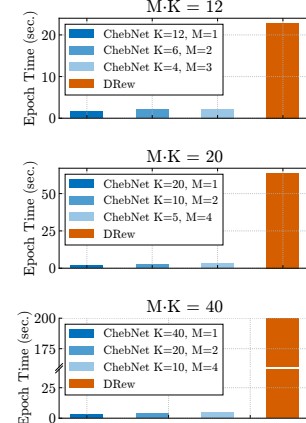

Figure 2: Epoch times for DRew and ChebNet for different receptive fields on the peptides-func task. M is the number of layers, and K is the number of filters.

### 2.3  The Connection between ChebNet and GCN

Although often interpreted as disparate models, GCN [57] can be derived as a special case of ChebNet [25] by truncating the Chebyshev expansion to $K = 1$ and making additional simplifications such as approximating $\tilde{\mathbf{L}} \approx \mathbf{I} - \mathbf{D}^{-1/2}\mathbf{A}\mathbf{D}^{-1/2}$ and adding self-loops to improve numerical stability. These choices introduced a strong locality bias, which aligned with widely used homophilic benchmarks and significantly reduced computational costs. Due to some of these strengths, GCN has become the de facto backbone for many modern GNNs, and has led to ChebNet being somewhat forgotten, under the assumption that it does not perform well and will not scale well in mid- to large-size graphs due to its spectral nature. For instance, until recently, ChebNet was almost never included in popular GNN benchmarks, such as [31], or in those designed to evaluate long-range dependencies, such as those in the Long Range Graph Benchmark

(LRGB) [33] and the numerous studies that leveraged this dataset to assess the effectiveness of graph rewiring, positional encodings, or Transformer-based architectures. ChebNet was similarly disregarded from large-scale graph evaluations such as the OGB benchmarks [55], presumably due to prevailing assumptions about its scalability.

# 3 Analyzing and Improving ChebNet from First Principles

## 3.1 The Effectiveness and Scalability of Vanilla ChebNet

In this subsection, we perform two high-level empirical tests to challenge the commonplace assumption that spectral GNNs inherently suffer from poor performance and limited scalability. We do so by firstly testing ChebNet [25] on a long-range test on the Ring Transfer dataset from [27], which has become a de facto benchmark for state-of-the-art methods seeking to model long-distance dependencies on graphs. Furthermore, to test scalability, we compare epoch training times on the `peptides-func` dataset from [33] with respect to state-of-the-art rewired MPNNs [49] based on a GCN backbone. We compare ChebNet with different numbers of filters $K$ and layers $M_{\text{Cheb}}$ with an $M_{\text{MPNN}}$-layer MPNN. To ensure a fair comparison, we ensure that $M_{\text{Cheb}}K = M_{\text{MPNN}}$ so that the two methods would have the same receptive field. The results are shown in Figures 2 and 3.

As seen from Figure 3, ChebNet is capable of performing long-range retrieval on the ring transfer dataset for rings of up to 50 nodes, which is a substantial improvement over a regular GCN. While this finding aligns with intuition, as we will formalize in the following sections, it may

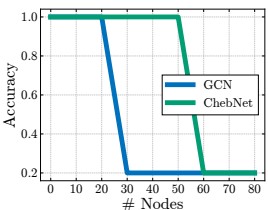

Figure 3: Test accuracy on RingTransfer.

nonetheless surprise practitioners, since ChebNet is not typically included as a benchmark in long-range studies such as that in [30]. On the other hand, as seen in Figure 2, ChebNet scales gracefully on standard tasks such as the `peptides-func` dataset. When compared to DRew, a state-of-the-art method that leverages both dynamic rewiring and delay to propagate information through a GCN (or other MPNN) backbone, we observe that DRew's epoch times are almost up to two orders of magnitude larger than ChebNet's. We note further that this inefficiency is not unique to this particular baseline: Graph Transformers incur quadratic scaling in the number of nodes and, by effectively discarding the underlying graph structure, trade-off inductive bias for increased compute, while many rewiring approaches depend on cubic-time algorithms (e.g., Floyd–Warshall or eigendecompositions). Together, these examples underscore how numerous contemporary techniques intended to overcome traditional message-passing limitations actually erode computational benefits, whereas ChebNet, a more natural spectral baseline that generalizes GCN, delivers both training speed and strong performance out-of-the-box.

## 3.2 Signal Propagation Analysis of ChebNet

In this subsection, we conduct a sensitivity analysis of ChebNet via the spectral norm of the Jacobian of node features, providing an exact characterization of ChebNet's information flow through different layers and between pairs of nodes, in the spirit of [3]. Specifically, we begin by analyzing the layer-wise Jacobian for Spectral GNNs that use polynomial filters. In this setting, we demonstrate that the layer-wise Jacobian becomes unstable as the polynomial order $K$ increases. Lastly, we investigate the sensitivity of node pairs when using ChebNet. We provide the proofs for the statements in Appendix B.

**Lemma 3.1** (Layer-Wise Jacobian for a Spectral GNN). *Consider a linear spectral GNN whose layer-wise update is performed through the following polynomial filter $f(\mathbf{X}) = \sum_{k=1}^{K} T_k(\mathbf{L}) \mathbf{X} \, \Theta_k$, where $\mathbf{X} \in \mathbb{R}^{n \times d}$ is the node feature matrix, $T_k(\mathbf{L}) \in \mathbb{R}^{n \times n}$ is the $k$-th polynomial of the Laplacian $\mathbf{L} \in \mathbb{R}^{n \times n}$, and $\Theta_k \in \mathbb{R}^{d \times d'}$ are learnable weight matrices. Then, the vectorized Jacobian $\mathbf{J} = \partial \operatorname{vec}(f(\mathbf{X})) / \partial \operatorname{vec}(\mathbf{X})$ is*

$$\mathbf{J} = \sum_{k=1}^{K} \Theta_k^{\top} \otimes T_k(\mathbf{L}). \tag{2}$$

Given the layer-wise Jacobian from Lemma 3.1, we proceed to analyze the dynamics of a Spectral GNN in Theorem 1 below. Specifically, we focus on the case where the polynomial filter can be approximated by powers of the Laplacian, i.e., $T_k(\mathbf{L}) = \mathbf{L}^k$.

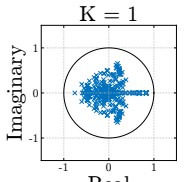 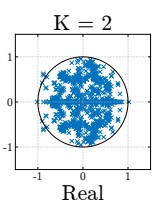 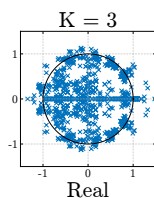 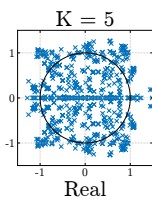 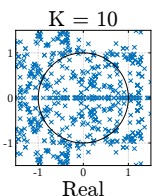

Figure 4: Singular-value spectra of the graph-wise Jacobian in the complex plane for vanilla ChebNet with increasing polynomial order $K$.

**Theorem 1** (Layer-Wise Jacobian singular-value distribution). *Assume the setting of Lemma 3.1, with $T_k(\mathbf{L}) = \mathbf{L}^k$ and $\mathbf{L}$ the symmetric normalized Laplacian, and let all $\mathbf{\Theta}_k \in \mathbb{R}^{d \times d}$ be initialized with i.i.d. $\mathcal{N}(0, \sigma^2)$ entries. Denote the eigenvalues of $\mathbf{L}$ as $\{\lambda_1, \ldots, \lambda_n\}$, the squared singular values of $\mathbf{\Theta}_k \mathbf{\Theta}_k^T$ as $\{\mu_{1,k}, \ldots, \mu_{d,k}\}$, and the squared singular values of the Jacobian by $\gamma_{i,j}$. Then, for sufficiently large $d$ the empirical eigenvalue distribution of $\mathbf{\Theta}_k \mathbf{\Theta}_k^T$ converges to the Marchenko-Pastur distribution $\forall k$. Then, the mean and variance of each $\gamma_{i,j}$ are*

$$\mathbb{E}\big[\gamma_{i,j}\big] = \sigma^2 \sum_{k=1}^{K} \lambda_i^{2k}, \tag{3}$$

$$\mathrm{Var}\big[\gamma_{i,j}\big] = \sigma^4 \left( \sum_{k=1}^{K} \lambda_i^{2k} \right)^2. \tag{4}$$

Theorem 1 shows that the singular values spectrum of the layer-wise Jacobian depends on the sum over the powers of the normalized Laplacian's eigenvalues. This indicates that larger polynomial orders $K$ push the singular value-spectrum towards unstable dynamics, as empirically demonstrated in Figure 4. Stacking several layers of large filter orders will therefore severely hinder the trainability of a Spectral GNN of this form.

Beyond analyzing the information propagation dynamics between different layers, we are interested in quantifying the communication ability between distant nodes in the graph. Recent literature has proposed to measure information flow in the graph by evaluating the sensitivity of a node embedding after $l$ layers (i.e., hops of propagation) with respect to the input of another node using the node-wise Jacobian [82, 27, 3], i.e., $\partial \mathbf{x}_u^{(l)} / \partial \mathbf{x}_v^{(0)}$. Following this approach, we measure how sensitive a node embedding of ChebNet at an arbitrary layer $l$ with respect to the initial features of another node, to better illustrate the long-range propagation capabilities of spectral GNNs.

**Theorem 2** (ChebNet Sensitivity). *Consider a Chebyshev-based Graph Neural Network (ChebNet) defined as:*

$$\mathbf{X}^{(l+1)} = \sum_{k=0}^{K} T_k(\mathbf{L}) \mathbf{X}^{(l)} \mathbf{W}_k^{(l)}, \tag{5}$$

*where $\mathbf{L} \in \mathbb{R}^{n \times n}$ is the graph Laplacian, $T_k(\mathbf{L})$ is the $k$-th Chebyshev polynomial of the Laplacian, and $\mathbf{W}_k^{(l)}$ are learnable weight matrices. Assume activation function $\sigma$ is identity and let $\mathbf{X}^{(0)}$ be the input features.*

*Then, the sensitivity of node $v$ with respect to node $u$ after $l$ layers is given by:*

$$\frac{\partial \mathbf{x}_v^{(l)}}{\partial \mathbf{x}_u^{(0)}} = \left( \prod_{l=0}^{l-1} \left( \sum_{k=0}^{K} T_k(\mathbf{L}) \mathbf{W}_k^{(l)} \right) \right)_{v,u}. \tag{6}$$

This result indicates that the sensitivity of ChebNet is closely tied to the polynomial order $K$. We note that, in the case of $K = 1$, the sensitivity aligns with that of standard MPNNs [1] (such as GCN), exhibiting reduced long-range communication. However, for $K > 1$, the higher polynomial orders significantly enhance the ChebNet's sensitivity, enabling more effective long-range propagation and improving the overall capacity to capture distant dependencies in the graph. Therefore, we conclude that a large filter order K is needed to enable long-range communication between nodes in the graph, but this will result in unstable training dynamics. This serves to explain the decay in performance in Figure 3. In the next subsection, we will propose a remedy to this issue.

---

[1] The sensitivity upper bound of standard MPNNs is further discussed in Appendix B.5.

### 3.3 Stable-ChebNet: Stability with Antisymmetric Parameterization

As discussed in Section 3.2, although ChebNet demonstrates strong long-range propagation capabilities, increasing the polynomial order can introduce significant instability into the model dynamics. To address this challenge, we propose Stable-ChebNet, a simple yet effective modification of classical ChebNet aimed at improving its stability. Recent literature has demonstrated that the effectiveness of neural architectures can be significantly improved by framing them as stable, non-dissipative dynamical systems [50, 18, 45, 46, 54]. The core idea behind these approaches is to carefully regulate the spectrum of the Jacobian matrix to ensure the network operates within a stable regime. Specifically, this behavior can be achieved by constraining the eigenvalues of the Jacobian to be purely imaginary. Under this constraint, the input graph information is effectively propagated through the successive transformations into the final nodes' representation. Motivated by this line of work, we begin by reformulating ChebNet as a continuous-time differential equation. Specifically, we consider the following ordinary differential equation (ODE):

$$\frac{d\mathbf{X}(t)}{dt} = \sum_{k=0}^{K} T_k(\mathbf{L})\mathbf{X}(t)\mathbf{W}_k \tag{7}$$

for time $t \in [0, T]$ and subject to the initial condition (i.e., the input features) $\mathbf{X}(0) = \mathbf{X}^{(0)}$. In other words, the dynamics of the system (i.e., the continuous flow of information over the graph) is now described as the ChebNet update rule. To ensure the Jacobian of this system has purely imaginary eigenvalues, a straightforward approach is to use antisymmetric weight matrices[2] and the symmetrically normalized Laplacian. This choice, as formalized in the following theorem, directly enforces the desired spectral property, leading to inherently stable dynamics.

**Theorem 3** (Purely Imaginary Eigenvalues). *Let $\mathbf{L}$ be the symmetric normalized Laplacian and $-\mathbf{W}_k = \mathbf{W}_k^\top$ $\forall k = 0, \ldots, K$, then the graph-wise Jacobian of the ODE in Equation (7) has purely imaginary eigenvalues, i.e.,*

$$Re(\lambda_i(\mathbf{J})) = 0, \ \forall i. \tag{8}$$

Even in this section, we provide the proofs for the statements in Appendix B. Theorem 3 shows that by enforcing antisymmetry in the weight matrices and leveraging the symmetric structure of the Laplacian, we guarantee that the Jacobian has purely imaginary eigenvalues, ensuring that node representations remain sensitive to input features of far away nodes without suffering from the instability of standard ChebNet.

As for standard differential-equation-inspired neural architectures, a numerical discretization method is needed to solve Equation (7). We solve the equation with a simple finite difference scheme, i.e., forward Euler's method, yielding the following node update equation

$$\mathbf{X}^{(l+1)} = \mathbf{X}^{(l)} + \epsilon \left( \sum_{k=0}^{K} T_k(\mathbf{L})\mathbf{X}^{(l)}(\mathbf{W}_k - \mathbf{W}_k^\top - \gamma\mathbf{I}) \right) \tag{9}$$

where $\mathbf{I}$ is the identity matrix, $\gamma \in \mathbb{R}$ is a hyper-parameter that maintains the stability of the forward Euler method, and $\epsilon \in \mathbb{R}_+$ is the discretization step.

We refer to the ODE in Equation (9) as Stable-ChebNet, and in the following, we show that this new formulation achieves second-order stability. This is a critical improvement, as a naive Euler discretization of the original ChebNet (without imposing constraints on the Jacobian eigenvalues) would result in only first-order stability, which is considerably more prone to numerical instability. Specifically, without these constraints, the model can exhibit exponential growth or decay in the gradients, significantly limiting its ability to capture long-range dependencies [3]. In contrast, second-order stable systems, like Stable-ChebNet, maintain controlled gradient dynamics over longer timescales, allowing for effective long-range information propagation.

---

[2]A matrix $\mathbf{A} \in \mathbb{R}^{d \times d}$ is antisymmetric (i.e., skew-symmetric) if $-\mathbf{A} = \mathbf{A}^\top$

Table 1: Mean test set $\log_{10}(\text{MSE})$ and standard deviation averaged over 4 random weight initializations for each configuration on the Graph Property Prediction dataset. The lower the better.

| Model | Diameter | SSSP | Eccentricity |
|---|---|---|---|
| GCN | $0.7424 \pm 0.0466$ | $0.9499 \pm 0.0001$ | $0.8468 \pm 0.0028$ |
| GAT | $0.8221 \pm 0.0752$ | $0.6951 \pm 0.1499$ | $0.7909 \pm 0.0222$ |
| GraphSAGE | $0.8645 \pm 0.0401$ | $0.2863 \pm 0.1843$ | $0.7863 \pm 0.0207$ |
| GIN | $0.6131 \pm 0.0990$ | $-0.5408 \pm 0.4193$ | $0.9504 \pm 0.0007$ |
| GCNII | $0.5287 \pm 0.0570$ | $-1.1329 \pm 0.0135$ | $0.7640 \pm 0.0355$ |
| DGC | $0.6028 \pm 0.0050$ | $-0.1483 \pm 0.0231$ | $0.8261 \pm 0.0032$ |
| GRAND | $0.6715 \pm 0.0490$ | $-0.0942 \pm 0.3897$ | $0.6602 \pm 0.1393$ |
| A-DGN w/ GCN backbone | $0.2271 \pm 0.0804$ | $-1.8288 \pm 0.0607$ | $0.7177 \pm 0.0345$ |
| ChebNet | $-0.1517 \pm 0.0343$ | $-1.8519 \pm 0.0539$ | $-1.2151 \pm 0.0852$ |
| **Stable-ChebNet (ours)** | $\mathbf{-0.2477 \pm 0.0526}$ | $\mathbf{-2.2111 \pm 0.0160}$ | $\mathbf{-2.1043 \pm 0.0766}$ |

**Theorem 4** (Non-exponential Information Growth or Decay with Antisymmetric Weights). *Consider the Stable-type Chebyshev Graph Neural Network (Stable-ChebNet) defined by:*

$$\mathbf{X}^{(l+1)} = \mathbf{X}^{(l)} + \epsilon \sum_{k=0}^{K} T_k(\mathbf{L})\mathbf{X}^{(l)}\mathbf{W}_k^{(l)}, \tag{10}$$

*with small step size $\epsilon > 0$ and antisymmetric weight matrices:*

$$(\mathbf{W}_k^{(l)})^\top = -\mathbf{W}_k^{(l)}, \quad \forall k, l. \tag{11}$$

*Then the Jacobian $J^{(l)}$ of the layer does not lead to exponential growth or decay across layers. Specifically, we have:*

$$\|\mathbf{J}^{(l)}\|_2 = 1 + O(\epsilon^2). \tag{12}$$

*Conversely, for general weights without the antisymmetric property, exponential growth or decay of the Jacobian norm typically occurs.*

# 4 Experiments

We evaluate ChebNet and its stable formulation (**Stable-ChebNet**) across a variety of settings to thoroughly assess long-range capabilities. In this section, we report and discuss the performance of both models on these benchmarks. We report additional experiments on heterophilic node classification tasks from [70] in Appendix F. We run our experiments on a single A100 GPU and provide the full details on the hyperparameter search for all datasets in Appendix D, and baseline and datasets details in Appendix C.

**Graph Property Prediction Dataset.** We evaluate our model's ability to predict long-range graph properties using a synthetic dataset developed by Corso et.al in [24] under the experimental setup of [45]. The dataset consists of undirected graphs drawn from a diverse set of random and structured families (Erdős–Rényi, Barabási–Albert, caterpillar, etc), ensuring a broad coverage of topological properties. Each graph contains between 25 and 35 nodes as per the setup of Gravina et.al in [45] (in contrast to the 15–25 node range originally used by [24]), thus increasing task complexity and raising the need for long-range information propagation. Finally, each node is assigned a single scalar feature sampled uniformly at random from the interval $[0, 1]$. We provide a detailed comparison in Table 1.

**Results.** Compared to classical ChebNet, Stable-ChebNet yields consistent and significant gains across all three tasks. On the *Diameter* task, classical ChebNet achieves a $\log_{10}(\text{MSE})$ of $-0.15$, whereas Stable-ChebNet improves it to $-0.25$. On the *Single Source Shortest Path (SSSP)* task, the gain is larger and goes from $-1.85$ using ChebNet to $-2.21$ with our Stable-ChebNet form. Finally, on *Eccentricity*, where the difficulty is highest due to the necessity to propagate information about the most distant nodes individually, classical ChebNet achieves a $\log_{10}(\text{MSE})$ of $-1.22$ while Stable-ChebNet reaches $-2.10$, reducing the average prediction error by more than an order of magnitude relative to the baseline. Relative to other baselines, Stable-ChebNet dominates most models based on standard message-passing or modified diffusion mechanisms. Methods like GCN and GCNII are badly over-squashed, barely reaching negative $\log_{10}(\text{MSE})$ values.

**Over-Squashing Analysis on Barbell Graphs.** To further investigate the robustness of our model to oversquashing, we use as a benchmark the barbell regression tasks introduced in [5]. In this task, a model that fails to transfer any information across the single bridge edge will produce an essentially random constant and obtain a mean-squared error (MSE) close to 1; an error in the 0.4–0.6 band indicates that only a partial amount of information has overcome the bottleneck. Errors around $\approx 0.25$ and below suggest that the oversquashing has been effectively overcome. In this work, we compare Stable-ChebNet's performance on barbell graphs of varying sizes ($N = 10, 25, 50, 100$) against numerous baselines, mainly an MLP and variants of MPNNs such as GCN [57], GAT [83], and SAGE [51]. Further description of the task is found in Appendix C.2.

**Results.** We observe in Figure 5 that both a classical ChebNet and Stable-ChebNet successfully learn the small $N = 10$ case with negligible error. However, for moderate graph sizes with $N = 50$, a classical ChebNet with fixed $K = 8$ already sits in the "partial collapse" regime as its MSE increases to around 0.90 and slides towards the random-guess area as $N$ keeps growing (Table 2). For the same range of hops K, replacing the standard update with our stable Euler-based formulation keeps the error almost two orders of magnitude smaller with an MSE below 0.20, confirming that the non-dissipative time-stepping effectively prevents the over-squashing phenomenon.

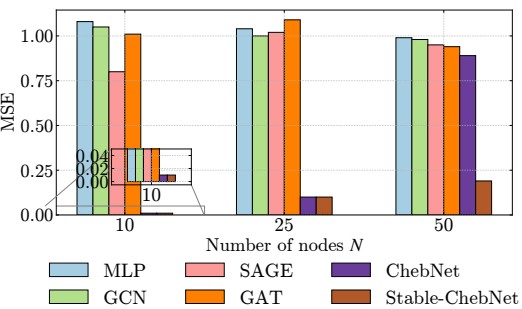

Figure 5: Mean Squared Error (MSE) comparison of various MPNN baselines at different node counts $N$ of Barbell graphs.

Table 2: Mean squared error (MSE) of ChebNet and Stable-ChebNet on the over-squashing experiment for barbell graphs. Left: sizes $N = 50, 70$ for $K = 9$ and 10. Right: size $N = 100$ for $K = 20$.

| Method | K | 50 | 70 |
|---|---|---|---|
| ChebNet | $K = 9$ | $0.32 \pm 0.39$ | $1.08 \pm 0.05$ |
| | $K = 10$ | $\mathbf{0.05 \pm 0.00}$ | $1.08 \pm 0.01$ |
| **Stable-ChebNet** | $K = 9$ | $0.17 \pm 0.11$ | $0.47 \pm 0.49$ |
| | $K = 10$ | $\mathbf{0.05 \pm 0.00}$ | $\mathbf{0.06 \pm 0.03}$ |

| Method | K | 100 |
|---|---|---|
| ChebNet | $K = 20$ | $0.87 \pm 0.05$ |
| **Stable-ChebNet (ours)** | $K = 20$ | $\mathbf{0.21 \pm 0.27}$ |

**Open-Graph Benchmark.** To evaluate real-world applicability on large-scale graphs, we run experiments for node-level tasks on two large-scale graph datasets from the Open-Graph Benchmark (OGB) [55]. `ogbn-arxiv` is a citation network in which each node corresponds to an academic paper, and the task is node classification by predicting the subject area of unseen papers. The other dataset we use, `ogbn-proteins`, is a protein–protein interaction network aimed at inferring protein functions. To ensure a fair comparison and emphasize efficiency, we limit the number of parameters in our models to be within the same range as those used in existing and most recent OGB benchmarks.

**Results.** Table 3 reports the performance on the `ogbn-arxiv` citation network, where ChebNet achieves around 73% test accuracy, while Stable-ChebNet boosts the performance further to 75.7%, outperforming all other methods, including a variety of MPNNs and Graph Transformers such as GraphGPS [71] and Exphormer [80]. Similarly, on the `ogbn-proteins` interaction network, ChebNet attains about 77.6% accuracy compared to Stable-ChebNet's 79.5% (Table 4). Competing approaches achieve nearly 72% for MPNN-based methods, while Transformer-based models' performances range from 77.4% for NodeFormer [88] to 79.5% for SGFormer [89]. Hence, Stable-ChebNet remarkably competes and often outperforms state-of-the-art models on this benchmark, demonstrating that the Euler formulation consistently narrows the gap with and in some cases overtakes Transformer baselines such as SGFormer [89] and Spexphormer [79]. Together, these results demonstrate that augmenting ChebNet with an Euler step not only addresses the classical ChebNet's shortcomings on long-range information propagation but also performs effectively well on graphs with hundreds of thousands of nodes, in contrast to regular-sized graphs seen in previous experiments.

Table 3: Accuracy on `ogbn-arxiv`.

| Model | ogbn-arxiv |
|---|---|
| GCN | $71.74 \pm 0.29$ |
| ChebNet | $73.27 \pm 0.23$ |
| ChebNetII | $72.32 \pm 0.23$ |
| GraphSAGE | $71.49 \pm 0.27$ |
| GAT | $72.01 \pm 0.20$ |
| NodeFormer | $59.90 \pm 0.42$ |
| GraphGPS | $70.92 \pm 0.04$ |
| GOAT | $72.41 \pm 0.40$ |
| EXPHORMER+GCN | $72.44 \pm 0.28$ |
| SPEXPHORMER | $70.82 \pm 0.24$ |
| **Stable-ChebNet (ours)** | $\mathbf{75.73 \pm 0.51}$ |

Table 4: Accuracy on `ogbn-proteins`.

| Model | ogbn-proteins |
|---|---|
| MLP | $72.04 \pm 0.48$ |
| GCN | $72.51 \pm 0.35$ |
| ChebNet | $77.55 \pm 0.43$ |
| SGC | $70.31 \pm 0.23$ |
| GCN-NSAMPLER | $73.51 \pm 1.31$ |
| GAT-NSAMPLER | $74.63 \pm 1.24$ |
| SIGN | $71.24 \pm 0.46$ |
| NodeFormer | $77.45 \pm 1.15$ |
| SGFormer | $79.53 \pm 0.38$ |
| **SPEXPHORMER** | $\mathbf{80.65 \pm 0.07}$ |
| Stable-ChebNet (ours) | $79.55 \pm 0.34$ |

**Long-Range Graph Benchmark (LRGB).** LRGB [33] is a collection of GNN benchmarks that evaluate models on tasks involving long-range interactions. We use two of its molecular-property datasets: Peptides-func for graph classification and Peptides-struct for graph regression.

**Results.** A detailed leaderboard is shown in Table 5. It can be seen that Stable-ChebNet improves upon its vanilla counterpart. Together with S2GCN, it reaches an average precision (AP) above 70 on Peptide-func and a Mean Absolute Error (MAE) below 0.26 on the regression task, bearing in mind that S2GCN requires a more expensive full Laplacian eigendecomposition. Overall, our model achieves competitive performance on peptide structures with results competing with and often outperforming some well-known graph-based models including graph transformers such as Exphormer [80] and GraphViT [53], state space models including Graph Mamba and GMN [84], and rewiring methods like DRew [49]. It is worth noting that the gain in AP for DRew comes at the cost of computing positional encodings (Laplacian eigenvectors) for every graph before training, while Stable-ChebNet does not use any positional encodings.

**Heterophilic benchmarks** We further assess Stable-ChebNet on node-classification tasks explicitly designed to stress performance under heterophily, following the standardized protocol of Platonov et al. ("Roman-empire", "Amazon-ratings", "Minesweeper" and "Tolokers") [70]. We keep the exact data processing, splits, and metrics recommended therein. Concretely, we report accuracy on Roman-empire and Amazon-ratings, and ROC-AUC on Minesweeper and Tolokers averaging over four random initializations as in the protocol.

**Link to oversmoothing, heterophily, and long-rangeness.** Our heterophilic results (Table 9) should not be over-interpreted as "evidence of long-range propagation" or as a direct antidote to oversmoothing. The recent position paper by Arnaiz-Rodríguez & Errica [2] argues that several widespread assumptions in the literature are often conflated: (i) that heterophily is inherently detrimental while homophily is beneficial, (ii) that long-range propagation is best evaluated on heterophilic graphs, and (iii) that performance degradation mainly arises from oversmoothing. They show that heterophily, long-range interactions, and oversmoothing are orthogonal factors: a graph may be heterophilic yet dominated by local dependencies, or homophilic yet require long-range reasoning. Hence, evaluations should focus on the nature of the learning task, not merely on global homophily ratios. In this light, our Stable-ChebNet scores on Roman-empire, Minesweeper, and Tolokers demonstrate that a stable spectral propagator has competitive performance on standardized heterophily benchmarks, but they do not by themselves certify long-rangeness. Those conclusions are better supported by our dedicated long-range tests and stability analysis.

## 5   Conclusion

In this work, we have re-examined ChebNet, one of the earliest spectral GNNs, from first principles, uncovering its innate ability to capture long-range dependencies via higher-order polynomial filters, but also its susceptibility to unstable propagation dynamics as the polynomial order grows. By casting ChebNet as a continuous-time ODE and imposing antisymmetric weight constraints, we introduced

Table 5: Long-range benchmark results. AP is the target metric on *peptides-func* (higher is better), and MAE is the target metric on *peptides-struct* (lower is better).

| Model Type | Model | peptides-func (AP ↑) | peptides-struct (MAE ↓) |
|---|---|---|---|
| *Transformer* | SAN+LapPE | $63.84 \pm 1.21$ | $0.2683 \pm 0.0043$ |
| | TIGT | $66.79 \pm 0.74$ | $0.2485 \pm 0.0015$ |
| | Specformer | $66.86 \pm 0.64$ | $0.2550 \pm 0.0014$ |
| | Exphormer | $65.27 \pm 0.43$ | $0.2481 \pm 0.0007$ |
| | G.MLPMixer | $69.21 \pm 0.54$ | $0.2475 \pm 0.0015$ |
| | Graph ViT | $69.42 \pm 0.75$ | $0.2449 \pm 0.0016$ |
| | GRIT | $69.88 \pm 0.82$ | $0.2460 \pm 0.0012$ |
| *Rewiring* | LASER | $64.40 \pm 0.10$ | $0.3043 \pm 0.0019$ |
| | DRew-GCN | $69.96 \pm 0.76$ | $0.2781 \pm 0.0028$ |
| | +PE | $71.50 \pm 0.44$ | $0.2536 \pm 0.0015$ |
| *State Space* | Graph Mamba | $67.39 \pm 0.87$ | $0.2478 \pm 0.0016$ |
| | GMN | $70.71 \pm 0.83$ | $0.2473 \pm 0.0025$ |
| | MP-SSM | $69.93 \pm 0.52$ | $0.2458 \pm 0.0017$ |
| *GNN* | A-DGN | $59.75 \pm 0.44$ | $0.2874 \pm 0.0021$ |
| | ChebNet | $69.61 \pm 0.33$ | $0.2627 \pm 0.0033$ |
| | ChebNetII | $68.19 \pm 0.27$ | $0.2618 \pm 0.0058$ |
| | GCN | $68.60 \pm 0.50$ | $0.2460 \pm 0.0007$ |
| | GRAMA | $70.93 \pm 0.78$ | $\mathbf{0.2436 \pm 0.0022}$ |
| | GRAND | $57.89 \pm 0.62$ | $0.3418 \pm 0.0015$ |
| | GraphCON | $60.22 \pm 0.68$ | $0.2778 \pm 0.0018$ |
| | PH-DGN | $70.12 \pm 0.45$ | $0.2465 \pm 0.0020$ |
| | SWAN | $67.51 \pm 0.39$ | $0.2485 \pm 0.0009$ |
| | PathNN | $68.16 \pm 0.26$ | $0.2545 \pm 0.0032$ |
| | CIN++ | $65.69 \pm 1.17$ | $0.2523 \pm 0.0013$ |
| | **S2GCN** | $\mathbf{72.75 \pm 0.66}$ | $\mathbf{0.2467 \pm 0.0019}$ |
| | +PE | $\mathbf{73.11 \pm 0.66}$ | $\mathbf{0.2447 \pm 0.0032}$ |
| | Stable-ChebNet (ours) | $70.32 \pm 0.26$ | $0.2542 \pm 0.0030$ |

Stable-ChebNet, whose forward Euler discretization yields non-dissipative, second-order–stable information flow without resorting to costly eigendecompositions, positional encodings, or graph rewiring. We provide a theoretical analysis of Stable-ChebNet showing purely imaginary Jacobian spectra and bounded layerwise sensitivity. We support the analysis with extensive experiments on a variety of synthetic and other long-range graph benchmarks. Stable-ChebNet consistently matches or outperforms state-of-the-art message-passing, rewiring, state-space, and transformer-based models , while retaining the fundamental properties of Chebyshev filters. Future work can focus on generalizing this ODE-based framework to broader spectral GNN families, whereby the stability analysis can be extended to other polynomial types and orthogonal bases.

**Impact Statement.** This work aims to advance the field of machine learning on graph-structured data which are abundant in the real world. There are many potential societal consequences of our work, none of which we feel must be specifically highlighted here.

**Code Availability** All code and datasets will be made publicly available at https://github.com/ahariri13/Stable-ChebNet to support reproducibility and future work.

## Acknowledgments and Disclosure of Funding

AG and DB acknowledge funding from EU-EIC EMERGE (Grant No. 101070918). XD acknowledges support from the Oxford-Man Institute of Quantitative Finance and EPSRC No. EP/T023333/1.

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

# A   Supplementary Related Work

Effective propagation and preservation of information on graphs remains a central challenge in deep learning on graphs, especially when long-range communication between nodes becomes fundamental for the downstream task [77]. GNNs usually rely on local neighborhood aggregation, which limits their capacity to capture interactions between distant nodes [1, 27] due to challenges such as over-smoothing [15, 69, 72] and over-squashing [1, 82, 27], which are linked to the problem of vanishing gradients [3]. Several techniques have been proposed to address this issue.

Graph rewiring techniques [39, 1, 4] modify the original topology, usually as a pre-processing step, with the aim of directly connect distant nodes and facilitate information flow. Rewiring methods can be broadly classified based on the type of information they leverage, including curvature metrics [1, 68, 36], effective resistance [9], random walks [7], spectral gap [56], and node features [49].

Similarly, Graph Transformers [76, 59, 91, 71] enable direct message passing between any pair of nodes via attention mechanisms, employing an attention mechanism on the entire graph. These models often incorporate positional encodings, such as Laplacian eigenvectors [30] or random-walk structural encodings (RWSE) [32], to encode graph structure. To reduce the quadratic complexity of the full attention mechanism, recent methods introduce methods such as sparse attention mechanisms [92, 23], Exphormer [80], and linear graph transformers [26].

Despite the success of graph rewiring methods and Graph Transformers, these approaches often introduce additional computational complexity due to the use of denser graph shift operators. An alternative strategy to enhance long-range propagation focuses on constraining the dynamics of the GNN to remain stable and non-dissipative, while maintaining the computational complexity of classical MPNNs. In this paradigm, the GNN is interpreted as a discretization of a differential equation, leveraging dynamical systems theory to maintain a constant rate of information flow between nodes. This behavior has been achieved either through antisymmetric weight parameterization [45, 46, 47, 48] or by exploiting port-Hamiltonian dynamics [54].

Recent works further explore long-range interactions in GNNs. In [6], Bamberger et al. formalize and measure interaction ranges in graph operators, while in [61] Liang et al. introduce a large-scale dataset and metric to quantify long-range dependencies. These studies complement existing approaches by providing tools to better evaluate and understand information propagation across distant nodes.

Other approaches include filtering messages in the information flow [37, 35], using a graph adaptive method based on a learnable ARMA framework [34], state space models [8, 16], or fractional power of the graph shift operator [64].

# B   Sensitivity Results

In this section, we provide the proofs for the statements in Section 3.2.

## B.1   Proof of Lemma 3.1

*Proof.* Applying vectorization to $f(\mathbf{X})$ and recalling that $\mathrm{vec}(\mathbf{AXB}) = (\mathbf{B}^\top \otimes \mathbf{A})\,\mathrm{vec}(\mathbf{X})$, we obtain

$$\mathrm{vec}(f(\mathbf{X})) = \sum_{k=1}^{K}(\mathbf{\Theta}_k^\top \otimes \mathbf{L}^k)\,\mathrm{vec}(\mathbf{X}).$$

Taking derivatives:

$$\mathbf{J}_f = \frac{\partial f}{\partial\,\mathrm{vec}(\mathbf{X})} = \sum_{k=1}^{K}(\mathbf{\Theta}_k^\top \otimes T_k(\mathbf{L})).$$

$\square$

## B.2   Proof of Theorem 1

*Proof.* Define the $k$-hop Jacobian block $\mathbf{J}_k = \mathbf{\Theta}_k \otimes \mathbf{L}^k$ and the full Jacobian $\mathbf{J} = \sum_{k=1}^{K}\mathbf{J}_k$. Because $\mathbf{\Theta}_k \otimes \mathbf{L}^k$ has singular values $s_j(\mathbf{\Theta}_k)\,|\lambda_i|^k$, the squared singular values of $\mathbf{J}_k$ are given by $\gamma_{i,j}^{(k)} = \lambda_i^{2k}\,\mu_{j,k}$.

For a *square* Gaussian matrix, the empirical spectrum of $\Theta_k\Theta_k^\top$ converges to the Marchenko–Pastur (MP) law. Scaling by $\sigma^2\lambda_i^{2k}$ yields

$$m_1^{(k)} = \sigma^2\lambda_i^{2k} \tag{13}$$

$$m_2^{(k)} = 2(\sigma^2\lambda_i^{2k})^2. \tag{14}$$

Independence, together with the rotational symmetry of each $\mathbf{\Theta}_k$ implies the blocks $\mathbf{J}_k$ are *asymptotically free*. Freeness gives additive $R$-transforms, hence additive *free* cumulants $\kappa_r\left(\sum_k \mathbf{J}_k\right) = \sum_k \kappa_r(\mathbf{J}_k)$. For $r = 1, 2$ the classical moments coincide with the free cumulants, so the ordinary moments of $\gamma_{i,j}$ also add:

$$m_r = \sum_{k=1}^{K} m_r^{(k)}, \qquad r \in \{1, 2\}. \tag{15}$$

Insert Equation (13) into Equation (15):

$$m_1 = \sigma^2 \sum_{k=1}^{K} \lambda_i^{2k} \tag{16}$$

$$m_2 = 2\sigma^4 \sum_{k=1}^{K} \lambda_i^{4k}. \tag{17}$$

Finally,

$$\begin{aligned}
Var[\gamma_{i,j}] &= m_2 - m_1^2 \\
&= 2\sigma^4 \sum_k \lambda_i^{4k} - \sigma^4 \left(\sum_k \lambda_i^{2k}\right)^2 \\
&= \sigma^4 \left(\sum_k \lambda_i^{2k}\right)^2.
\end{aligned}$$

$\square$

### B.3 Proof of Theorem 2

*Proof.* Recall that the forward pass for one ChebNet layer (omitting the activation function) is:

$$\mathbf{X}^{(l+1)} = \sum_{k=0}^{K} T_k(\mathbf{L})\mathbf{X}^{(l)}\mathbf{W}_k^{(l)}. \tag{18}$$

Applying vectorization, we obtain:

$$\mathrm{vec}(\mathbf{X}^{(l+1)}) = \sum_{k=0}^{K} \left((\mathbf{W}_k^{(l)})^\top \otimes T_k(\mathbf{L})\right) \mathrm{vec}(\mathbf{X}^{(l)}). \tag{19}$$

By unrolling Equation (19) and taking the derivative with respect to $\mathrm{vec}(\mathbf{X}^{(0)})$, we obtain the sensitivity after $l$ layers:

$$\frac{\partial \mathrm{vec}(\mathbf{X}^{(l)})}{\partial \mathrm{vec}(\mathbf{X}^{(0)})} = \prod_{l=0}^{l-1}\left(\sum_{k=0}^{K} \left((\mathbf{W}_k^{(l)})^\top \otimes T_k(\mathbf{L})\right)\right). \tag{20}$$

Focusing on a single feature channel (or summing across channels), we obtain the sensitivity:

$$\frac{\partial \mathbf{X}^{(l)}}{\partial \mathbf{X}^{(0)}} = \prod_{l=0}^{l-1}\left(\sum_{k=0}^{K} T_k(\mathbf{L})\mathbf{W}_k^{(l)}\right). \tag{21}$$

Then, the sensitivity of node $v$ with respect to node $u$ after $l$ layers is given by:

$$\frac{\partial \mathbf{x}_v^{(l)}}{\partial \mathbf{x}_u^{(0)}} = \left(\prod_{l=0}^{l-1}\left(\sum_{k=0}^{K} T_k(\mathbf{L})\mathbf{W}_k^{(l)}\right)\right)_{v,u}. \tag{22}$$

$\square$

## B.4 Proof of Theorem 3

*Proof.* Given that the graph-wise Jacobian of Equation (7) is Equation (2), we note that each term in the Jacobian, of the form $\mathbf{W}_k^\top \otimes T_k(\mathbf{L})$, is an antisymmetric matrix. This follows from the fact that $\mathbf{W}_k$ is antisymmetric by construction, $T_k(\mathbf{L})$ preserves the symmetry of the normalized Laplacian, and the Kronecker product of an antisymmetric matrix with a symmetric matrix is itself antisymmetric. Finally, since the sum of antisymmetric matrices remains antisymmetric, it follows that the graph-wise Jacobian of Equation (7) has purely imaginary eigenvalues. $\square$

## B.5 Sensitivity upperbound of standard MPNNs

In the following theorem we report the sensitivity upperbound computed for standard MPNNs in [27].

**Theorem 5** (Sensitivity uppperbound of standard MPNNs, taken from [27]). *Consider a standard MPNN with $l$ layers, where $c_\sigma$ is the Lipschitz constant of the activation $\sigma$, $w$ is the maximal entry-value over all weight matrices, and $d$ is the embedding dimension. For $u, v \in V$ we have*

$$\left\| \frac{\partial \mathbf{h}_v^{(l)}}{\partial \mathbf{h}_u^{(0)}} \right\| \leq \underbrace{(c_\sigma w d)^l}_{model} \underbrace{(\mathbf{O}^l)_{vu}}_{topology}, \tag{23}$$

*with $\mathbf{O} = c_r \mathbf{I} + c_a \mathbf{A} \in \mathbb{R}^{n \times n}$ is the message passing matrix adopted by the MPNN, with $c_r$ and $c_a$ are the contributions of the self-connection and aggregation term.*

## B.6 Proof of Theorem 4

*Proof.* First, recall the Jacobian explicitly:

$$\mathbf{J}^{(l)} = \mathbf{I} + \epsilon \sum_{k=0}^{K} \left( (\mathbf{W}_k^{(l)})^\top \otimes T_k(\mathbf{L}) \right). \tag{24}$$

Define:

$$A^{(l)} = \sum_{k=0}^{K} \left( (\mathbf{W}_k^{(l)})^\top \otimes T_k(\mathbf{L}) \right). \tag{25}$$

**Antisymmetric Case:** When $(\mathbf{W}_k^{(l)})^\top = -\mathbf{W}_k^{(l)}$, the matrix $A^{(l)}$ is antisymmetric, because it is a Kronecker product of antisymmetric and symmetric matrices. Its eigenvalues are purely imaginary (or symmetric about zero), meaning their real parts vanish. Thus, for the spectral radius, we have:

$$\|\mathbf{J}^{(l)}\|_2^2 = \rho \left( (\mathbf{J}^{(l)})^\top \mathbf{J}^{(l)} \right) \tag{26}$$

$$= \rho \left( \mathbf{I} + \epsilon (A^{(l)} + (A^{(l)})^\top) + \epsilon^2 (A^{(l)})^\top A^{(l)} \right). \tag{27}$$

Due to antisymmetry:

$$(A^{(l)})^\top + A^{(l)} = 0. \tag{28}$$

Hence:

$$\|\mathbf{J}^{(l)}\|_2^2 = \rho \left( \mathbf{I} + \epsilon^2 (A^{(l)})^\top A^{(l)} \right). \tag{29}$$

The matrix $(A^{(l)})^\top A^{(l)}$ is symmetric and positive semi-definite, having nonnegative real eigenvalues. Expanding around the identity gives:

$$\|\mathbf{J}^{(l)}\|_2 = \sqrt{1 + \epsilon^2 \lambda_{\max} \left( (A^{(l)})^\top A^{(l)} \right)} = 1 + O(\epsilon^2). \tag{30}$$

Thus, no exponential growth or decay occurs.

**General Case of Stable-ChebNet (Without Antisymmetry):** For arbitrary matrices, the linear term $(A^{(l)} + (A^{(l)})^\top)$ typically does not vanish, introducing nonzero real eigenvalues. This causes exponential growth or decay in the Jacobian norm across layers:

$$\|\mathbf{J}^{(l)}\|_2 \approx 1 \pm C\epsilon, \tag{31}$$

for some constant $C = \max_{i=0,\cdots,l} C^{(i)} > 0$. Iterating over layers results in exponential instability:

$$\|\mathbf{J}^{(l)}\mathbf{J}^{(l-1)}\ldots\mathbf{J}^{(0)}\|_2 \approx (1 \pm C\epsilon)^l, \tag{32}$$

which grows or decays exponentially as the depth $l$ increases.

Thus, antisymmetric weights provide explicit protection against exponential growth or decay, while general weights typically do not.

**Standard ChebNet Case:** Iterating over layers, this results in exponential instability:

$$\|\mathbf{J}^{(l)}\mathbf{J}^{(l-1)}\ldots\mathbf{J}^{(0)}\|_2 \approx (\pm C)^l, \tag{33}$$

which grows or decays exponentially as the depth $l$ increases.

$\square$

## C  Dataset and Baseline Description

### C.1  Graph Property Prediction task description

For the graph property prediction dataset, we make use of three separate tasks:

**Diameter (Graph-Level).** The diameter is defined as the length of the longest shortest path between any two nodes in the graph. It requires aggregating information from distant regions of the graph.

**SSSP (Node-Level).** Single-Source Shortest Path requires predicting each node's distance to a designated source node. Solving this task with GNNs places a strong emphasis on propagating information from nodes that may lie many hops away from the source.

**Eccentricity (Node-Level).** For each node $u$, the eccentricity is the length of the maximum shortest path between $u$ and any other node. As with diameter and SSSP, accurate eccentricity estimation relies heavily on capturing long-distance relationships.

In total, the task contains 5,120 graphs for training, 640 for validation, and 1,280 for testing. The train/validation/test splits follow the same seed and setup in [45]. We train our models by optimizing mean squared error (MSE) for each task, performing a grid search over hyperparameters (e.g., learning rate, weight decay, number of layers). Each experiment is repeated across four random initializations, and we report the average performance.

Because shortest-path-related tasks inherently rely on propagating signals over large portions of a graph, they serve as a natural stress test for the capacity of GNNs to perform long-range propagation.

### C.2  Barbell graph task description

A Barbell graph $B_{n,k}$ is formed by connecting two complete graphs $K_n$ (the "bells") with a simple path of length $k$ (the "bridge"). Therefore, every node inside a bell has high intra-cluster connectivity $(n-1$ neighbors), whereas the bridge nodes have degree 2 and constitute the only communication route between the two bells. In this task, the target for nodes is to output the average input feature over nodes of the opposite bell and vice-versa, as illustrated in Figure 6. **Mean squared error (MSE)** is used for node-level regression as a proxy for how severely the GNN is either *oversquashing* (failing to pass information across the narrow "bridge") or *oversmoothing* (collapsing all node embeddings to be nearly identical). Numerical MSE outcomes are associated with each pathology.

**An MSE of around 1 corresponds to oversquashing:** The model is so bottlenecked by the bridge that it effectively ignores or fails to incorporate information from the other "bell." In other words, each side of the barbell can predict its own node labels, but the information from the opposite side never gets through, leading to a characteristic level of error ($\approx 1$ in their chosen label distribution).

**An MSE of around 30 corresponds to oversmoothing:** The model passes messages so many times (or in such a way) that it "collapses" all node embeddings toward the same prediction, ignoring local distinctions within each side. Because the barbell's node labels are diverse (randomly assigned), forcing every node toward the same value yields a much larger overall MSE ($\approx 30$ in their setup).

On the other hand, an MSE of around 0.5 as shown in the results simply means that the model is doing better than the severe oversquashing case (it is not completely failing to pass information across

the barbell). In other words, some amount of meaningful communication is happening between the two "bells," and the node representations are not entirely collapsed.

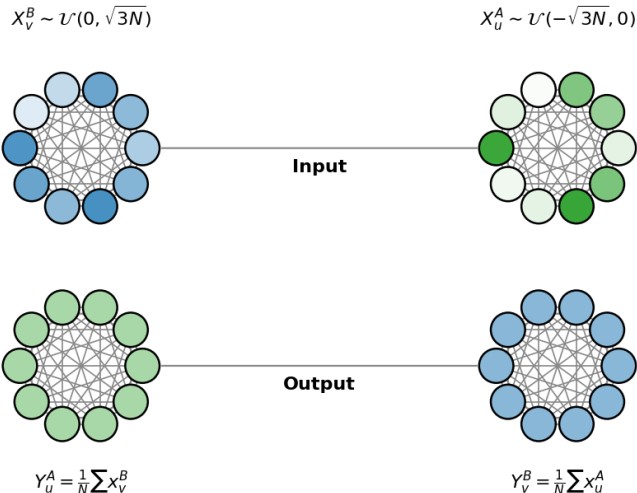

Figure 6: Illustration of input and output node features in a barbell graph setup. Adapted from Bamberger et.al [5].

## C.3 Employed baselines

In our experiments, the performance of our method is compared with various state-of-the-art GNN baselines from the literature. Specifically, we consider:

- Classical GNN methods, i.e., GCN [57], GraphSAGE [51], GAT [83], GIN [90], and GC-NII [20], ChebNet [25], ChebNetII [52], CIN++ [42], PathNN [66], S2GCN [40], SGC [87], SIGN [38], GRAMA [34], GatedGCN [13], CoGNN [37], H2GCN [95], CPGNN [94], FAGCN [11], GPR-GNN [21], FSGNN [65], GloGNN [60], GBK-GNN [29], JacobiConv [85];

- Differential-equation inspired GNNs (DE-GNNs), i.e., DGC [86], GRAND [17], Graph-CON [73], A-DGN [45], and SWAN [46] PH-DGN [54];

- Graph Transformers, i.e., SAN [59], GraphGPS [71], GOAT [58], Exphormer [80], GRIT[63], GraphViT [53], G.MLPMixer [53] SPEXPHORMER [79], TIGT [22], SG-Former [89], NodeFormer [88], Specformer [10], GCN-nsampler and GAT-nsampler [93],

  GT [78], NAGphormer [19], Polynormer [26];

- Rewiring-based methods, i.e., LASER [7], and DRew [49];

- SSM-based GNN, i.e., Graph-Mamba [84], GMN [8], MP-SSM [16]

# D   Hyper-parameter grids

Tables 6 to 8 summarize our hyperparameter exploration: Table 6 listing the sweep ranges for Stable-ChebNet on Peptides-func, graph-property benchmarks (Diameter, SSSP, Eccentricity), and ogbn-arxiv along with ogbn-proteins experiments, respectively.

Table 6: Hyper-parameter grid for Stable-ChebNet ablation on *Peptides-func*.

| Hyper-parameter | Reference | Sweep values used in ablation |
|---|---|---|
| Hidden dim $d$ | **140** | 100, 120, 140, 145, 160 |
| Polynomial order $K$ | **10** | 6,8,10 |
| Num of layers | **4** | 3,4,5 |
| MLP layers | **2** | 1,2,3 |
| Step size $\varepsilon$ | **0.5** | $[0.1, 1.0]$ |
| Dissipative force $\gamma$ | **0.05** | 0.001, 0.01, 0.05, 0.1 |
| Batch size | **64** | 32, 64, 128 |
| Learning rate | **0.001** | $0.0001, 0.001, 0.01$ |
| Optimizer | AdamW | AdamW |
| Pos-enc type | None | None, Laplacian, RW |
| Pos-enc dim | **16** | 8, 16, 32 |

Table 7: Hyper-parameter grid and best settings for Stable-ChebNet on three synthetic graph-property benchmarks.

| Hyper-parameter | Values in grid | Diam | SSSP | Ecc |
|---|---|---|---|---|
| Hidden dimension $d$ | 20, 30, 50 | 50 | 30 | 30 |
| Number of layers | 1, 2, 3, 5, 10, 20 | 20 | 5 | 5 |
| Polynomial order $K$ | 3, 5, 10 | 4 | 10 | 10 |
| Step size $\varepsilon$ | 0.01, 0.10, 0.20, 0.30 | 0.40 | 0.30 | 0.30 |
| Dissipative force $\gamma$ | 0, 0.01, 0.50, 1 | 0.01 | 0.00 | 0.00 |
| Activation function | `tanh, relu` | `relu` | `relu` | `relu` |
| Learning rate | 0.001,0.003 | 0.003 | 0.003 | 0.003 |
| Weight decay | $1 \times 10^{-6}$ | $1 \times 10^{-6}$ | $1 \times 10^{-6}$ | $1 \times 10^{-6}$ |

Table 8: Hyper-parameter sweep ranges for Stable-ChebNet on `ogbn-arxiv` and `ogbn-proteins`.

| Hyper-parameter | Sweep (arxiv) | Sweep (proteins) |
|---|---|---|
| Hidden dim $d$ | 128, 256, 512 | 256, 512, 1024 |
| Polynomial order $K$ | 4, 5, 6, 10 | 5, 10, 15 |
| Num of layers | 2, 3, 4, 5 | 3, 5, 7 |
| MLP layers | 1, 2, 3 | 1, 2, 3 |
| Step size $\epsilon$ | $[0.1, 1.0]$ | $[0.1, 1.0]$ |
| Dissipative force $\gamma$ | 0.01, 0.05, 0.1 | 0.01, 0.05, 0.1 |
| Batch size | 256, 512, 1024 | 512, 1024, 2048 |
| Learning rate | 0.001, 0.01, 0.05 | 0.0005, 0.001, 0.005 |
| Optimizer | Adam | Adam |
| Pos-enc type | None, Laplacian, RW | None, Laplacian, RW |
| Pos-enc dim | 8, 16, 32 | 16, 32, 64 |

# E  Eigenvalues distribution comparison

Figures 7a and 7b provide an empirical comparison of the Jacobian eigenvalue spectra for a classical ChebNet versus our Stable-ChebNet layer at $K = 5$. The top two panels illustrate ChebNet's spectrum: the left histogram shows that both the real and imaginary parts of its eigenvalues span broadly from roughly $-2$ to $+2$, with pronounced peaks near the extremes signaling a large spectral radius and a susceptibility to unstable, oscillatory dynamics. The accompanying scatter plot maps these eigenvalues in the complex plane, revealing many points lying well outside the unit circle, which corroborates the theoretical prediction that high-order Chebyshev filters can push the system toward chaotic regimes.

In contrast, the bottom row portrays the spectrum of Stable-ChebNet. Its histogram (bottom left) is tightly concentrated within approximately $-0.4$ to $+0.4$ on both axes, indicating that eigenvalues

remain far inside the unit circle. The complex-plane scatter (bottom right) further demonstrates that all eigenvalues lie symmetrically about the imaginary axis and are bounded in magnitude, consistent with the antisymmetric weight parameterization, guaranteeing purely imaginary Jacobian eigenvalues and therefore more stable dynamics.

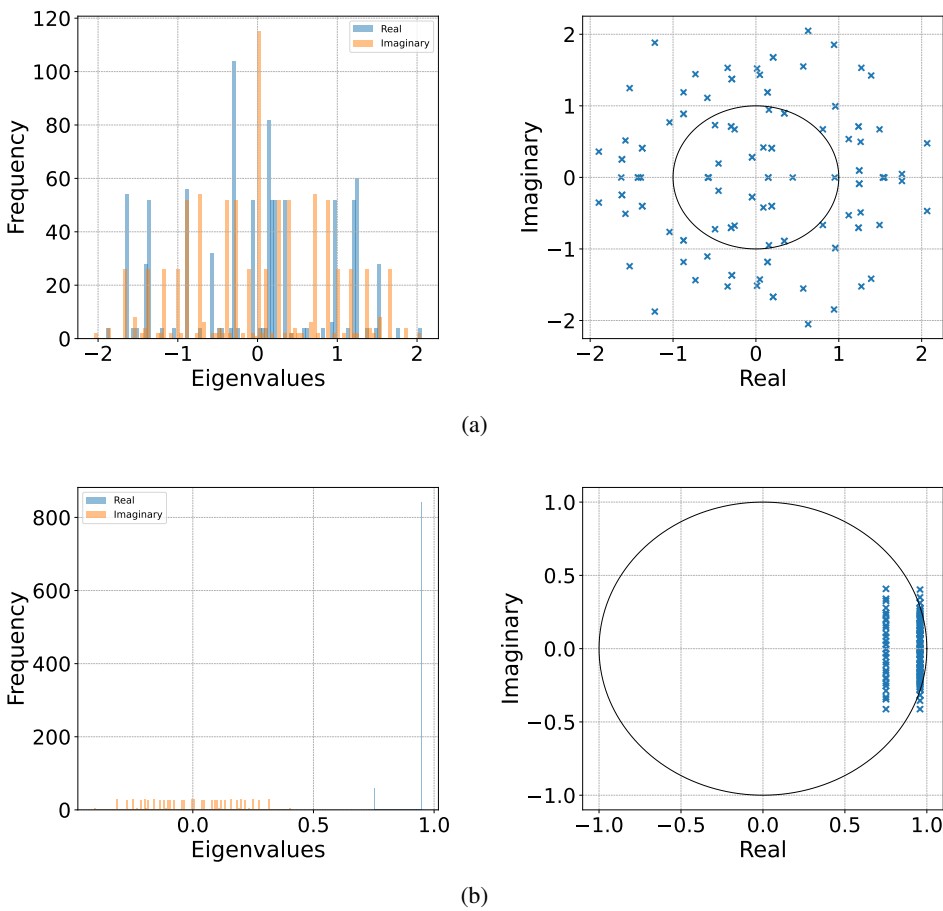

Figure 7: Comparison of eigenvalue distributions: (a) ChebNet and (b) StableChebNet.

## F   Additional Experiments on Heterophilic Benchmarks

To further evaluate the performance of our Stable-ChebNet, we assess its the effectiveness in capturing complex relational information in heterophilic settings, where nodes belonging to same class are often connected through longer and sparser paths, we consider the five node classification tasks introduced in [70]. Specifically, we consider the "Roman-empire", "Amazon-ratings" and "Minesweeper", "Tolokers" datasets. We adhere to the same data and experimental setting presented in [70].

Table 9: Mean test set score and std averaged over 4 random weight initializations on heterophilic datasets. The higher, the better.

| Model | Roman-empire Acc ↑ | Amazon-ratings Acc ↑ | Minesweeper AUC ↑ | Tolokers AUC ↑ |
|---|---|---|---|---|
| **[62]** | | | | |
| MLP-2 | $66.04_{\pm0.71}$ | $49.55_{\pm0.81}$ | $50.92_{\pm1.25}$ | $74.58_{\pm0.75}$ |
| SGC-1 | $44.60_{\pm0.52}$ | $40.69_{\pm0.42}$ | $82.04_{\pm0.77}$ | $73.80_{\pm1.35}$ |
| MLP-1 | $64.12_{\pm0.61}$ | $38.60_{\pm0.41}$ | $50.59_{\pm0.83}$ | $71.89_{\pm0.82}$ |
| **MPNNs** | | | | |
| GAT | $80.87_{\pm0.30}$ | $49.09_{\pm0.63}$ | $92.01_{\pm0.68}$ | $83.70_{\pm0.47}$ |
| GAT (LapPE) | $84.80_{\pm0.46}$ | $44.90_{\pm0.73}$ | $93.50_{\pm0.54}$ | $84.99_{\pm0.54}$ |
| GAT (RWSE) | $86.62_{\pm0.53}$ | $48.58_{\pm0.41}$ | $92.53_{\pm0.65}$ | $85.02_{\pm0.67}$ |
| Gated-GCN | $74.46_{\pm0.54}$ | $43.00_{\pm0.32}$ | $87.54_{\pm1.22}$ | $77.31_{\pm1.14}$ |
| GCN | $73.69_{\pm0.74}$ | $48.70_{\pm0.63}$ | $89.75_{\pm0.52}$ | $83.64_{\pm0.67}$ |
| GCN (LapPE) | $83.37_{\pm0.55}$ | $44.35_{\pm0.36}$ | $94.26_{\pm0.49}$ | $84.95_{\pm0.78}$ |
| GCN (RWSE) | $84.84_{\pm0.55}$ | $46.40_{\pm0.55}$ | $93.84_{\pm0.48}$ | $85.11_{\pm0.77}$ |
| CO-GNN($\Sigma$, $\Sigma$) | $91.57_{\pm0.32}$ | $51.28_{\pm0.56}$ | $95.09_{\pm1.18}$ | $83.36_{\pm0.89}$ |
| CO-GNN($\mu$, $\mu$) | $91.37_{\pm0.35}$ | $54.17_{\pm0.37}$ | $97.31_{\pm0.41}$ | $84.45_{\pm1.17}$ |
| SAGE | $85.74_{\pm0.67}$ | $53.63_{\pm0.39}$ | $93.51_{\pm0.57}$ | $82.43_{\pm0.44}$ |
| **Graph Transformers** | | | | |
| Exphormer | $89.03_{\pm0.37}$ | $53.51_{\pm0.46}$ | $90.74_{\pm0.53}$ | $83.77_{\pm0.78}$ |
| NAGphormer | $74.34_{\pm0.77}$ | $51.26_{\pm0.72}$ | $84.19_{\pm0.66}$ | $78.32_{\pm0.95}$ |
| GOAT | $71.59_{\pm1.25}$ | $44.61_{\pm0.50}$ | $81.09_{\pm1.02}$ | $83.11_{\pm1.04}$ |
| GPS | $82.00_{\pm0.61}$ | $53.10_{\pm0.42}$ | $90.63_{\pm0.67}$ | $83.71_{\pm0.48}$ |
| GPS$_{\text{GCN+Performer}}$ (LapPE) | $83.96_{\pm0.53}$ | $48.20_{\pm0.67}$ | $93.85_{\pm0.41}$ | $84.72_{\pm0.77}$ |
| GPS$_{\text{GCN+Performer}}$ (RWSE) | $84.72_{\pm0.65}$ | $48.08_{\pm0.85}$ | $92.88_{\pm0.50}$ | $84.81_{\pm0.86}$ |
| GPS$_{\text{GCN+Transformer}}$ (LapPE) | OOM | OOM | $91.82_{\pm0.41}$ | $83.51_{\pm0.93}$ |
| GPS$_{\text{GCN+Transformer}}$ (RWSE) | OOM | OOM | $91.17_{\pm0.51}$ | $83.53_{\pm1.06}$ |
| GT | $86.51_{\pm0.73}$ | $51.17_{\pm0.66}$ | $91.85_{\pm0.76}$ | $83.23_{\pm0.64}$ |
| GT-sep | $87.32_{\pm0.39}$ | $52.18_{\pm0.80}$ | $92.29_{\pm0.47}$ | $82.52_{\pm0.92}$ |
| Polynormer | $92.55_{\pm0.30}$ | $\mathbf{54.81}_{\pm0.49}$ | $97.46_{\pm0.36}$ | $85.91_{\pm0.74}$ |
| **Heterophily-Designated GNNs** | | | | |
| CPGNN | $63.96_{\pm0.62}$ | $39.79_{\pm0.77}$ | $52.03_{\pm5.46}$ | $73.36_{\pm1.01}$ |
| FAGCN | $65.22_{\pm0.56}$ | $44.12_{\pm0.30}$ | $88.17_{\pm0.73}$ | $77.75_{\pm1.05}$ |
| FSGNN | $79.92_{\pm0.56}$ | $52.74_{\pm0.83}$ | $90.08_{\pm0.70}$ | $82.76_{\pm0.61}$ |
| GBK-GNN | $74.57_{\pm0.47}$ | $45.98_{\pm0.71}$ | $90.85_{\pm0.58}$ | $81.01_{\pm0.67}$ |
| GloGNN | $59.63_{\pm0.69}$ | $36.89_{\pm0.14}$ | $51.08_{\pm1.23}$ | $73.39_{\pm1.17}$ |
| GPR-GNN | $64.85_{\pm0.27}$ | $44.88_{\pm0.34}$ | $86.24_{\pm0.61}$ | $72.94_{\pm0.97}$ |
| H2GCN | $60.11_{\pm0.52}$ | $36.47_{\pm0.23}$ | $89.71_{\pm0.31}$ | $73.35_{\pm1.01}$ |
| JacobiConv | $71.14_{\pm0.42}$ | $43.55_{\pm0.48}$ | $89.66_{\pm0.40}$ | $68.66_{\pm0.65}$ |
| **Graph SSMs** | | | | |
| GMN | $87.69_{\pm0.50}$ | $54.07_{\pm0.31}$ | $91.01_{\pm0.23}$ | $84.52_{\pm0.21}$ |
| GPS + Mamba | $83.10_{\pm0.28}$ | $45.13_{\pm0.97}$ | $89.93_{\pm0.54}$ | $83.70_{\pm1.05}$ |
| GRAMA$_{\text{GCN}}$ | $88.61_{\pm0.43}$ | $53.48_{\pm0.62}$ | $95.27_{\pm0.71}$ | $\mathbf{86.23}_{\pm1.10}$ |
| MP-SSM | $90.91_{\pm0.48}$ | $53.65_{\pm0.71}$ | $95.33_{\pm0.72}$ | $85.26_{\pm0.93}$ |
| **Ours** | | | | |
| Stable-ChebNet | $\mathbf{92.03}_{\pm0.85}$ | $53.15_{\pm0.21}$ | $\mathbf{95.71}_{\pm2.26}$ | $85.55_{\pm3.35}$ |

