# OpenReview forum: "Return of ChebNet: Understanding and Improving an Overlooked GNN on Long Range Tasks"
_NeurIPS.cc/2025/Conference — NeurIPS 2025 spotlight_

### Official Review · Reviewer_XzyD · 2025-06-18

**Clarity:** 3
**Significance:** 3
**Originality:** 3
**Rating:** 4
**Confidence:** 5

**Summary:**

This paper revisits ChebNet, a classical spectral Graph Neural Network (GNN), and proposes a stabilized variant, Stable-ChebNet, for long-range graph tasks. The authors analyze the instability of high-order Chebyshev polynomial filters using a continuous-time ODE interpretation and propose antisymmetric weight parameterizations with dissipative terms to mitigate gradient explosion. Empirical evaluations on long-range benchmarks such as LRGB and OGB show that Stable-ChebNet achieves strong performance, competitive with or exceeding attention-based graph Transformers and rewiring techniques. While the approach is grounded in classical spectral theory, it is presented with modern theoretical and empirical rigor.

**Questions:**

- How does Stable-ChebNet compare in performance and scalability to recent models using hierarchical self-attention [1], multiresolution message passing [2, 3] or virtual node [4]?

- Would integrating learnable hierarchies or dynamic clustering (as in adaptive attention or pooling-based GNNs) [2] further enhance ChebNet’s capacity for long-range dependency modeling?

- Could the antisymmetric formulation be extended to more expressive spectral bases such as graph wavelets or multiresolution spectral domains [5, 6]?

References:

[1] Thuan Nguyen Anh Trang, Khang Nhat Ngo, Hugo Sonnery, Thieu Vo, Siamak Ravanbakhsh, Truong-Son Hy, Scalable Hierarchical Self-Attention with Learnable Hierarchy for Long-Range Interactions, Transactions on Machine Learning Research (TMLR). URL: https://openreview.net/pdf?id=qH4YFMyhce

[2] Truong-Son Hy and Risi Kondor, Multiresolution Equivariant Graph Variational Autoencoder, Machine Learning: Science and Technology, Volume 4, Number 1, DOI 10.1088/2632-2153/acc0d8. URL: https://iopscience.iop.org/article/10.1088/2632-2153/acc0d8

[3] Nhat Khang Ngo, Truong-Son Hy, and Risi Kondor, Multiresolution Graph Transformers and Wavelet Positional Encoding for Learning Long-Range and Hierarchical Structures, Journal of Chemical Physics, Volume 159, Issue 3, DOI 10.1063/5.0152833. URL: https://pubs.aip.org/aip/jcp/article/159/3/034109/2903066/Multiresolution-graph-transformers-and-wavelet

[4] Chen Cai, Truong-Son Hy, Rose Yu, and Yusu Wang, On the Connection Between MPNN and Graph Transformer, ICML 2023, Proceedings of Machine Learning Research 202:3408-3430. URL: https://proceedings.mlr.press/v202/cai23b/cai23b.pdf

[5] Truong-Son Hy and Risi Kondor, Multiresolution Matrix Factorization and Wavelet Networks on Graphs, ICML 2022 (Workshop on Topology, Algebra, and Geometry in Machine Learning), Proceedings of Machine Learning Research 196:172-182. URL: https://proceedings.mlr.press/v196/hy22a/hy22a.pdf

[6] Truong-Son Hy, Viet Bach Nguyen, Long Tran-Thanh and Risi Kondor, Temporal Multiresolution Graph Neural Networks For Epidemic Prediction, ICML 2022 (Workshop on Healthcare AI and COVID-19), Proceedings of Machine Learning Research 184:21-32. URL: https://proceedings.mlr.press/v184/hy22a/hy22a.pdf

**Ethical Concerns:**

["NO or VERY MINOR ethics concerns only"]

**Final Justification:**

I have read the authors' rebuttal as well as other reviewers' comments. I appreciate that the authors have conducted more experimental results based on my suggestions. I decided to keep my original rating as borderline accept.

**Limitations:**

- The proposed approach focuses narrowly on stabilizing ChebNet rather than expanding its expressiveness. By contrast, several recent works have introduced architectural innovations for long-range modeling (please see the references above).

- Stable-ChebNet assumes a fixed graph structure and does not address dynamic graphs or graphs with evolving topologies.

- The long-range benchmarks used still rely on synthetic or curated datasets; performance in realistic, noisy, or application-specific domains (e.g., biological networks or scientific simulations) is untested.

**Quality:**

3

**Strengths And Weaknesses:**

Strengths:
- Rigorous theoretical motivation, interpreting ChebNet as a dynamical system and identifying stability issues.

- Efficient implementation that avoids heavy architectural modifications or positional encodings.

- Solid benchmark results on long-range tasks.

- Reintroduces a classical method with modern improvements, offering a computationally lighter alternative to Transformers.

Weaknesses:
- The work does not sufficiently engage with recent advances in long-range modeling using adaptive attention and multiscale architectures such as hierarchical self-attention [1], multiresolution message passing [2], multiresolution graph transformers [3], and virtual node [4].

- The analysis assumes linear activations and simplified training dynamics, which may not reflect real-world performance under nonlinearity and optimization noise.

- There is no in-depth ablation to isolate the benefits of antisymmetric weights and the dissipative term.

References:

[1] Thuan Nguyen Anh Trang, Khang Nhat Ngo, Hugo Sonnery, Thieu Vo, Siamak Ravanbakhsh, Truong-Son Hy, Scalable Hierarchical Self-Attention with Learnable Hierarchy for Long-Range Interactions, Transactions on Machine Learning Research (TMLR). URL: https://openreview.net/pdf?id=qH4YFMyhce

[2] Truong-Son Hy and Risi Kondor, Multiresolution Equivariant Graph Variational Autoencoder, Machine Learning: Science and Technology, Volume 4, Number 1, DOI 10.1088/2632-2153/acc0d8. URL: https://iopscience.iop.org/article/10.1088/2632-2153/acc0d8

[3] Nhat Khang Ngo, Truong-Son Hy, and Risi Kondor, Multiresolution Graph Transformers and Wavelet Positional Encoding for Learning Long-Range and Hierarchical Structures, Journal of Chemical Physics, Volume 159, Issue 3, DOI 10.1063/5.0152833. URL: https://pubs.aip.org/aip/jcp/article/159/3/034109/2903066/Multiresolution-graph-transformers-and-wavelet

[4] Chen Cai, Truong-Son Hy, Rose Yu, and Yusu Wang, On the Connection Between MPNN and Graph Transformer, ICML 2023, Proceedings of Machine Learning Research 202:3408-3430. URL: https://proceedings.mlr.press/v202/cai23b/cai23b.pdf

---

> ### Author Rebuttal · Authors · 2025-07-31
>
> We thank the reviewer for raising these insightful and technically relevant questions. We appreciate the opportunity to clarify the positioning of Stable-ChebNet with respect to recent advancements in hierarchical and attention-based GNN architectures. Below, we provide a detailed discussion on the comparative aspects regarding performance, scalability, and potential extensions of our model through integration with learnable hierarchies or dynamic clustering mechanisms.
>
> **Performance comparison:**  The common dataset we share with the mentioned references is the Peptides dataset of Long Range Graph Benchmark (LRGB). Below we provide a comparative analysis that we would be happy to add in a revised version, especially given that Multi-Resolution Graph Transformers and Wavelet Positional Encodings make interesting benchmarks for our case.
>
> In terms of performance, StableChebNet has an overall comparative performance to the Multi-Resolution Graph Transformer [3], as the latter shows a better performance in the regression task of Peptides-struc, while our model’s advantage is in the classification task of Peptides-func.
>
> | Model             | Peptides-func (AP ↑) | Peptides-struct (MAE ↓) |
> |------------------|----------------------|--------------------------|
> | Sequoia + RWPE   | 0.6755 ± 0.0074       | **0.2453 ± 0.0006**      |
> | Sequoia + LapPE  | 0.6323 ± 0.0042       | 0.2526 ± 0.0019          |
> | MGT + LapPE      | 0.6728 ± 0.0152       | 0.2488 ± 0.0014          |
> | MGT + RWPE       | 0.6709 ± 0.0083       | 0.2496 ± 0.0009          |
> | MGT + WavePE     | 0.6817 ± 0.0064       | **0.2453 ± 0.0025**      |
> | Stable-ChebNet   | **0.7032 ± 0.0026**   | 0.2542 ± 0.0030          |
>
> Moreover, in our earlier experiments with the Peptides-func graph-level task, introducing a Virtual Node did not enhance our model’s performance, nor did the inclusion of positional encodings. We believe a potential reason for that is the fact that Stable-ChebNet and its vanilla counterpart already account well enough for the graph’s structure and cover a wide-enough receptive field that makes a global Virtual Node more redundant. We report the findings in the Table below, where we notice that a VN is more valuable for a GCN than for Stable ChebNet.
>
> | Model                  | Accuracy on Peptide-Func      |
> |------------------------|-------------------------------|
> | GCN                    | 68.60 ± 0.50                  |
> | GCN + VN               | 69.21 ± 0.25                  |
> |  Stable-ChebNet                | 70.32 ± 0.26                  |
> | Stable-ChebNet + VN    | 69.73 ± 0.33                  |
>
> **Dynamic clustering and scalability:** While making an interesting direction to explore, it is worth noting that Dynamic Clustering methods come with a certain computational cost, whether using adaptive attention or pooling-based GNNs that use Gumbel-Softmax trick for sampling, which might make the choice less plausible.  That being said, in terms of scalability, Stable‑ChebNet offers some advantages over the aforementioned approaches.
> Hierarchical attention models could incur O(nlog⁡n) complexity to compute global or multi‑scale attention maps, while Stable‑ChebNet performs only Chebyshev recurrences, yielding strictly O(K∣E∣) work per layer (with K the polynomial order). Similarly, the Multi-Resolution Graph Transformer performs costly precomputations: wavelet bases or Laplacian eigendecompositions that can scale as O(n^3) or require significant storage across scales, whereas our method needs only Chebyshev recurrence approximation the Laplacian eigendecomposition with repeated sparse multiplications.
> We make a similar point in our response to Reviewer E5Qi, highlighting that, unlike benchmarks such as DRew or S2GCN, our approach avoids expensive pre-processing steps for finding geodesic distances or performing full Laplacian eigendecomposition. Instead, we demonstrate the efficiency of using the Chebyshev approximation as a computationally lightweight alternative, with Stable-ChebNet offering stronger experimental results due to its non-dissipative properties.
>
> **Integrating learnable hierarchies:**  We believe Dynamic Clustering integrations can boost StableChebNet’s performance in task-specific scenarios. One example is graph-level tasks requiring both local AND global context where a notable amount of graph information is “lost” in pooling layers. The integration of hierarchical learning and Dynamic clustering  can be especially beneficial for very large graphs. In that scenario, a suitable approach consists of partitioning a very large graph (as in OGB benchmarks) and performing an **intra then inter-cluster stable propagation**. This strategy benefits from both advantages StableChebNet and dynamic clustering have to offer. For smaller graphs GCN could serve as a better backbone prior to dynamically clustering and learning in hierarchy, to allow for a fair combination of local and global information. This can be seen in the Table above whereby for the smaller Peptide graphs, Virtual Nodes, being one layer of hierarchical learning, had more benefit on top of GCN as compared to Stable-ChebNet which has already accounted for the global picture.
>
> On the other hand, many node-level tasks require very long-range propagation but do not necessarily benefit from a full global context vector. In that case, dynamic clustering might be counterproductive given the limited improvement and the higher cost of training associated with it. That being said, we believe that the benefit of hierarchical learning and the inclusion of global context vectors prior to classification highly depends on the task under disposal whereas the amount of local and global information needed should be tested. However, we believe this is indeed an interesting direction that should be explored in future extensions.
>
> **On studying temporal graphs :** We thank the reviewer for the insightful suggestion. Indeed handling dynamic and evolving networks is an important challenge in graph learning, especially considering its significance in numerous applications. While Stable-ChebNet focuses on the dynamics within the spatial domain, we agree that integrating it to the temporal setting would be a fascinating direction. Given the broad literature on temporal and dynamic GNNs, and the separate considerations for long-range dependencies in the temporal setting (See [7]) , we believe this would be more suitable for future work.
>
> **Further validation on a newly released benchmark:** We broaden our evaluation to include the recently released City-Networks benchmark by Liang et al. [8], which features large-scale, transductive tasks grounded in real-world urban road networks. This benchmark is specifically designed to assess GNNs' capacity to model long-range dependencies.
>
> The benchmarking results of that paper align with our findings and show (though without detailed analysis) that ChebNet performs well on long-range interactions given that it is often the best baseline and has a consistently high influence score, even on Ogbn-arxiv which we report in our paper. To further validate our model on real-world road networks, we ran our Stable-ChebNet on the Paris City-Networks data and compared it to vanilla ChebNet in the Table below.
>
> | # of Layers (L) | K (Hops) | ChebNet (%)      | Stable-ChebNet (%)  |
> |-----------------|----------|------------------|---------------------|
> | 2               | 10       | 39.96 ± 0.04     | 49.52 ± 0.12        |
> | 3               | 10       | 47.16 ± 0.08     | 51.22 ± 0.07        |
> | 3               | 12       | 47.39 ± 0.12     | 51.28 ± 0.28        |
>
> These results highlight Stable‑ChebNet’s effectiveness in capturing long-range dependencies across both synthetic and large-scale real-world scenarios, all while maintaining parameter efficiency.
>
> **On separating anti-symmetry:**  Following the reviewer’s suggestion, we carried out an additional experiment on the OGB‑Proteins benchmark to examine the isolated impact of antisymmetric weights in our model. We tested a variant that enforces antisymmetry but excludes the forward‑Euler discretization. This “antisymmetry‑only” model achieved a notably lower test accuracy (72.2%) compared to the full Stable‑ChebNet (79.5%), emphasizing the crucial role of combining antisymmetry with the discretization scheme for effective learning.
>
> | Model Variant           | Test Accuracy on OGB-Proteins |
> |-------------------------|-------------------------------|
> | Full Stable-ChebNet     | 79.5%                         |
> | ChebNet-Antisymmetry Only       | 72.2%                         |
>
> **Extension to other spectral bases:** Thank you for the thoughtful question. In this study, we chose to focus on ChebNet given its historical importance and its status as one of the most prominent and widely adopted spectral GNN models. While we anticipate that similar dynamics would arise with alternative bases such as graph wavelets and multiresolution spectral domains, formally extending our theoretical analysis to those settings would require a fresh examination of the spectral norm and Jacobian behavior. Providing such a comprehensive analysis is beyond the current scope but represents a promising avenue for future research.
>
> We genuinely hope that our responses and revisions have resolved your concerns and led you to reconsider your evaluation and score. We once again thank the reviewer for their valuable time and thoughtful assessment.
>
> [7] Marisca et al. (2025). Oversquashing in Spatiotemporal Graph Neural Networks . arXiv preprint https://arxiv.org/abs/2506.15507
>
> [8] Liang et al. (2025). Towards Quantifying Long-Range Interactions in Graph ML. arXiv preprint https://arxiv.org/abs/2503.09008

---

> > ### Comment · Reviewer_XzyD · 2025-08-05
> > **Acknowledgement**
> >
> > Thanks to the authors for addressing my main concern about the lacking of experimental results and comparison with recent baselines. I keep my original score (borderline accept).

---

> > > ### Author Response · Authors · 2025-08-09
> > >
> > > We thank the reviewer for their valuable feedback and engagement in the rebuttal process.

---

### Official Review · Reviewer_EiU1 · 2025-06-27

**Clarity:** 4
**Significance:** 4
**Originality:** 3
**Rating:** 5
**Confidence:** 5

**Summary:**

The authors note that the early ChebNet spectral GNN has surprisingly good capabilities for modeling long-range interactions on graphs. They revisit ChebNet and uncover that it suffers from an unstable behaviour during training. The authors are therefore proposing stable variants of ChebNet, that obtain competitive and state-of-the-art results on their empirical evaluation.

**Questions:**

- Please see the "Weaknesses" sections above.

- Did the authors also consider performing experiments with positional encodings (such as LapPE or RWSE)? The computation overhead is minimal, and I wonder if they could obtain further performance gains with this. This could also be an interesting and "low-hanging" addition to Table 5, since peptides-func normally benefits from positional encodings.

**Ethical Concerns:**

["NO or VERY MINOR ethics concerns only"]

**Final Justification:**

The authors have significantly improved their experimental evaluation section by adding new real-world and synthetic datasets. Moreover, the authors were very active during the rebuttal and have responded well to the concerns raised by all reviewers.

**Limitations:**

yes

**Quality:**

3

**Strengths And Weaknesses:**

**Strengths**:

1. The paper is well-written and easy to follow.
2. The paper continues a recent line of research where spectral models are reassessed for their effectiveness in modeling long-range interactions, with a notable related example being the Spatio-Spectral GNN [[1]]. I believe that this line of work (and the broader analysis of the behaviour of signals and weights of GNNs) is an important and promising direction going forward.
3. To my knowledge, this is the first paper that re-evaluates the performance of an overlooked model like ChebNet and demonstrates its competitiveness on tasks that are difficult for message-passing GNNs.
4. The design choices behind Stable-ChebNet are principled and well-justified. The authors also connect their approach to neural networks inspired by differential equations and provide theoretical support.
5. Stable-ChebNet performs well compared to MPNNs, Graph Transformers, and rewiring-based methods, particularly given that it does not rely on positional encodings.

**Weaknesses**:

1. The greatest weakness is that the empirical evaluation section is somewhat lacking. The paper would have been stronger if datasets that are not considered "long-range" were included, or even other datasets from the LRGB (such as PCQM-contact). Moreover, some datasets that are common when discussing long-range capabilities and over-squashing are also omitted (such as NeighborsMatch from [[2]]). The node-level datasets are also lacking, with only ogbn-proteins and ogbn-arxiv being used in the empirical evaluation. Since Stable-ChebNet improves signal propagation, I am wondering if it could also obtain better performance on heterophilic datasets, such as the ones introduced in [[3]].
2. Figure 1 is confusing and not very informative, especially in the context of the introduction. I suggest the authors clarify the figure's purpose and consider moving it to a later section.
3. The related work discussion is not fully fleshed out. Although Section 2 ("Background") mentions prior work, the coverage of spectral methods, rewiring strategies, and Graph Transformers is limited. The paper also shares conceptual connections with recent sequential and state-space models, such as [[4]], which explore similar ideas from a signal processing perspective. I recommend expanding the related work section in the appendix and, if space allows, in the main text.

**Conclusion:**

Overall, I believe that this is a good paper and should be accepted. However, my initial score is 4 (borderline accept). I will update my score to a 5 (accept) if the authors expand their empirical evaluation and address some of the other weaknesses I raised. Please note that I will not lower my score if the additional experiments turn out to be weak, since negative results would also be very informative, and the paper currently contains a good amount of strong results. I believe that the paper, the core idea, and the direction of research are strong, even if some results do not reach state-of-the-art performance. That said, the current empirical results are not sufficient for a practitioner or researcher to form a clear understanding of the model's overall performance.

I may further update my score depending on the comments and opinions of the other reviewers, particularly if they raise points that I strongly agree with or that significantly influence my assessment.


[1]: https://arxiv.org/abs/2405.19121
[2]:https://arxiv.org/abs/2006.05205
[3]:https://arxiv.org/abs/2302.11640
[4]:https://arxiv.org/pdf/2303.06349

---

> ### Author Rebuttal · Authors · 2025-07-31
>
> Thank you for the thoughtful and encouraging feedback. We appreciate the recognition of our clear writing, the importance of revisiting spectral methods for long-range interactions, and our novel re-evaluation of ChebNet. We're grateful for the acknowledgment of our principled design of Stable-ChebNet—its theoretical grounding, simplicity, and strong empirical performance without positional encodings. Your insights have been valuable in refining the paper. Below, we address your comments and outline our revisions.
>
> We thank the reviewer for their thoughtful feedback and valuable suggestions regarding the empirical evaluation section.
>
> **On node-level tasks:** We would like to clarify that, although it may not have been sufficiently emphasized in the original submission, our evaluation already includes several node-level prediction tasks. Specifically, the synthetic tasks—SSSP, Eccentricity, and the Barbell task—are formulated as node-level tasks, where the goal is to predict scalar values at each node. The only graph-level benchmarks in our evaluation are LRGB and the Diameter dataset in the GraphProp tasks, which we included to complement the node-level analyses.
>
> **Tree-NeighborsMatch dataset:** In response to the reviewer’s helpful suggestion, we have now included results on the Tree-NeighborsMatch, a dataset commonly used to assess long-range information propagation and over-squashing [2]. Up until a depth of 4, we can reach an accuracy of 100% with both Stable ChebNet and vanilla ChebNet. At depth 5, we reach 65% with vanilla ChebNet alone, which is already better than most baselines. Given the timeline of the rebuttal and the intense runtime at depths higher than 4, we were not able to run until depth 8 of this dataset, but we are happy to include them in the revised version. We expect that around depth 5 or 6, ChebNet will collapse while Stable ChebNet will persist as in the case of the Barbell task.
>
> **On the inclusion of heterophilic datasets:** We appreciate the suggestion to explore how label heterophily connects with long‑range dependency. In this work, however, we did not include standard heterophilic benchmarks, since there is currently no established correspondence between a graph’s heterophily and the need for long‑range information propagation—the latter being our primary focus. In fact, a recent position paper [5] argues that heterophily and long‑range reasoning are not inherently linked: the authors give examples of graphs that may exhibit strong heterophily yet only demand local feature aggregation, or conversely be highly homophilic but require messages to traverse many hops.
>
> To address your point empirically, we have now retrained both ChebNet and our Stable‑ChebNet model on the Roman‑Empire and Amazon‑Ratings datasets, using the hyperparameter settings from Platonov et al. [3]. Encouragingly, both models reach performance on par with the best updated MPNN results reported in the paper, while noting that the hyperparameter tuning was not extensive and it is possible that we could reach even higher results. An interesting observation from the tables below is the fact that even at a higher receptive field K, the degradation of Stable-ChebNet’s signal is less than that of ChebNet, as can be seen at K=6 for the Roman Empire dataset.
>
> **Dataset: Amazon-Ratings**
> | K | L | Stable-ChebNet |
> |---|---|---------------|
> | 4 | 4 | 53.80 ± 0.32  |
> | 3 | 4 | 54.17 ± 0.11  |
> | 3 | 5 | 54.24 ± 0.23  |
>
> **Dataset: Roman-Empire**
> |  K  |  L  |     ChebNet      |       Stable-ChebNet       |
> |:---:|:---:|:----------------:|:------------------:|
> |  3  |  7  |  91.26 ± 0.36    |  90.65 ± 1.08      |
> |  5  |  7  |  90.79 ± 1.19    |  89.77 ± 1.06      |
> |  6  |  7  |  84.75 ± 0.35    |  89.03 ± 2.19      |
>
> **Further validation on a newly released benchmark:** We extend our evaluation to the recently published City‑Networks benchmark by Liang et al. [4], which comprises large-scale, transductive tasks based on real-world urban road networks and is specifically designed to test GNNs’ ability to capture long-range dependencies.
> The benchmarking results of that paper align with our findings and show (though without detailed analysis) that ChebNet performs well on long-range interactions given that it is often the best baseline and has a consistently high influence score, even on Ogbn-arxiv which we report in our paper. To further validate our model on real-world road networks, we ran our Stable-ChebNet on the Paris City-Networks data and compared it to vanilla ChebNet in the Table below.
>
> | # of Layers (L) | K (Hops) | ChebNet (%)      | Stable-ChebNet (%)  |
> |-----------------|----------|------------------|---------------------|
> | 2               | 10       | 39.96 ± 0.04     | 49.52 ± 0.12        |
> | 3               | 10       | 47.16 ± 0.08     | 51.22 ± 0.07        |
> | 3               | 12       | 47.39 ± 0.12     | 51.28 ± 0.28        |
>
> These results highlight Stable‑ChebNet’s effectiveness in capturing long-range dependencies across both synthetic and large-scale real-world scenarios, all while maintaining parameter efficiency.
>
> **Clarifying Figure 1:** We apologize for the confusion surrounding Figure 1. This figure shows, at each layer, the "temperature" of each node—that is, the magnitude (or "heat") of its signal after being processed by the filter. In the classical ChebNet (top row), high-order Chebyshev filters behave like a dissipative diffusion process: by step T, most nodes have nearly "cooled" to zero, meaning that their signals have diminished significantly. This indicates that information dissipates over layers, leading to unbounded dynamics where small perturbations may grow uncontrollably or vanish. In contrast, our Stable ChebNet (bottom row) uses an antisymmetric, non-dissipative update rule that preserves signal magnitude even at large T. As a result, information remains stable and well-conditioned throughout the layers. We acknowledge that "temperature" was not clearly defined in the original caption and will include this explanation in the revised version.
>
> **Expanding the Related Work Discussion:** We appreciate the reviewer’s suggestion and will substantially enrich our discussion of rewiring strategies and Graph Transformers in the revision. In the main text’s Background section, we will introduce a broader spectrum of rewiring techniques and transformer-based graph architectures, highlighting how each approach addresses long-range dependency challenges. In particular, we will emphasize methods such as DRew for which we already include comparative runs alongside the benchmarked baselines in our tables. Several relevant transformer-based models are also represented in our main tables, including SPEXFormer, Exphormer, TIGT, and SAN+LapPE. These models reflect a range of strategies for overcoming long-range dependency and over-squashing issues in graph learning, and we will more clearly articulate how our approach relates to them. A more comprehensive comparison will be provided in the appendix, with key insights integrated into Section 2 where space allows.
>
> **Effect of Positional Encoding (and Virtual Node):** We did in fact extend our peptide‑function experiments with both Laplacian positional encodings (LapPE) and random‑walk structural encodings (RWSE), but found that neither led to gains—instead, accuracy dipped by roughly 1–2 points and per‑epoch training time increased by 20–30 percent. We suspect two main factors: first, ChebNet’s polynomial filters already capture much of the same spectral information that LapPE explicitly provides, so appending handcrafted eigenvector features can introduce redundancy or even noise, leading to slight overfitting on small peptide graphs. Second, computing LapPE via partial eigendecomposition (and RWSE via truncated walk statistics) incurs nontrivial overhead—especially on batches of variable‑size peptide graphs—so the resulting slowdown outweighs any marginal benefit. Given these findings, we opted not to include positional encodings in our main table, but we will add a brief note in Table 5’s caption summarizing that neither LapPE nor RWSE improved peptide‑function performance and that both incurred noticeable computational cost.
>
> Furthermore, in our preliminary experiments on the Peptides-func graph-level task, incorporating a Virtual Node did not lead to improved model performance, nor did the addition of positional encodings. We attribute this to the fact that both Stable-ChebNet and its vanilla version already capture the graph structure effectively and possess a sufficiently large receptive field, rendering the global Virtual Node largely redundant. The results, presented in the table below, show that the Virtual Node offers more benefit to GCN than to Stable-ChebNet.
>
> | Model                  | Accuracy on Peptide-Func      |
> |------------------------|-------------------------------|
> | GCN                    | 68.60 ± 0.50                  |
> | GCN + VN               | 69.21 ± 0.25                  |
> |  Stable-ChebNet                | 70.32 ± 0.26                  |
> | Stable-ChebNet + VN    | 69.73 ± 0.33                  |
>
> We sincerely hope that our clarifications and revisions have addressed your concerns and prompted a reconsideration of your evaluation and score. We thank again the reviewer for their time and consideration.
>
> [4] Liang et al. (2025). Towards Quantifying Long-Range Interactions in Graph ML. arXiv preprint https://arxiv.org/abs/2503.09008
>
> [5] Adrián Arnaiz‑Rodríguez and Federico Errica (2025). Oversmoothing, “Oversquashing”, Heterophily, Long‑Range, and more: Demystifying Common Beliefs in Graph Machine Learning. Position paper published on https://arxiv.org/abs/2505.15547

---

> ### Comment · Reviewer_EiU1 · 2025-08-05
>
> I thank the authors for their response.
>
> First, I would like to clarify that my suggestion regarding experiments on node-level tasks was not motivated by an interest in probing long-range capabilities, but rather by the goal of obtaining a more complete picture of the capabilities of (Stable-)ChebNet on these tasks:
>
> > The node-level datasets are also lacking, with only ogbn-proteins and ogbn-arxiv being used in the empirical evaluation. Since Stable-ChebNet improves signal propagation, I am wondering if it could also obtain better performance on heterophilic datasets, such as the ones introduced in [Platonov et al.];
>
> Regarding the authors’ comments on node-level datasets:
>
> > our evaluation already includes several node-level prediction tasks. Specifically, the synthetic tasks
>
> While these are indeed node-level tasks, I believe they are not a good proxy for real-world node-level applications. As the authors note, these are synthetic datasets primarily used to evaluate long-range capabilities. I maintain my view that ogbn-proteins and ogbn-arxiv are insufficient for a thorough real-world assessment of the proposed method. Additionally, the authors do not clarify why datasets such as PCQM-Contact from the LRGB benchmark are not included.
>
> That said, I appreciate the inclusion of the Trees, Roman-Empire, Amazon-Ratings, and City-Networks datasets. I believe the experimental evaluation section is now in better shape. Furthermore, the discussion on the use of positional encodings is interesting and adds value.
>
> At this point, I will maintain my score while following the ongoing discussion between the authors and Reviewer E5Qi, who I believe has similar concerns, and is raising some good points. I would like to observe the rest of the exchange before making a final decision.

---

> > ### Author Response · Authors · 2025-08-06
> >
> > We thank the reviewer for their response, their constructive feedback, and their willingness to further improve our work.
> >
> > Regarding the inclusion of PCQM-Contact, we would like to clarify that it was not considered in our evaluation primarily due to concerns raised by Tonshoff et al. [6], who highlighted certain limitations in the dataset’s metric design, specifically, challenges with handling false negatives and the inclusion of self-loops. These issues can affect model rankings and pose difficulties for fair benchmarking. For this reason, we included only the two peptide-related tasks. Nevertheless, to ensure full coverage on LRGB, we are willing to add PCQM in the next revision and report Stable-ChebNet’s performance there alongside Peptides.
> >
> > That said, we’re pleased the reviewer appreciated the inclusion of the new datasets. We agree that the experimental section has improved significantly as a result. In particular, we added node-level tasks during the rebuttal to enable a more comprehensive comparison. The tabulated results now highlight Stable-ChebNet’s performance across a range of benchmarks, covering varying levels of homophily and long-range dependencies.
> >
> > As suggested, we will incorporate the relevant discussion from the rebuttal into the revised version of the paper. Finally, we invite the reviewer to check the updated discussion with **Reviewer E5Qi**.
> >
> > We sincerely thank the reviewer once again for the thoughtful dialogue and genuinely hope that our responses and revisions have addressed your concerns and prompted you to reconsider your evaluation and score.
> >
> > [6] Tönshoff et.al. “Where Did the Gap Go? Reassessing the Long-Range Graph Benchmark.” Preprint, https://arxiv.org/pdf/2309.00367

---

> ### Comment · Reviewer_EiU1 · 2025-08-08
>
> I thank the authors for their rebuttal and discussions. They have done a very good job addressing most of the concerns, and I now believe that the paper is in a very good shape; therefore, I will recommend acceptance.

---

> > ### Author Response · Authors · 2025-08-09
> >
> > We are sincerely grateful for the reviewer's constructive feedback and their engagement in the rebuttal process.

---

### Official Review · Reviewer_zDKz · 2025-06-30

**Clarity:** 4
**Significance:** 2
**Originality:** 2
**Rating:** 5
**Confidence:** 4

**Summary:**

The paper shows the limitations of the ChebNet framework in propagating information in long range graphs. Specifically, the authors show that as the order of the chebyshev polynomial grows the response function becomes unstable (eigenvalues of the Jacobian exponentially increases). In order to circumvent this issue, the authors propose to restrict the weight matrices to be antisymmetric so as to have imaginary eigenvalues for the Jacobian. This bounds the norm of the Jacobian. In the modified method (Stable-Chebnet) the authors cast the Chebnet as a continuous ordinary differential equation and the layerwise update is obtained using the forward euler method. The method performs comparably to some SoTA methods on synthetic and long range benchmarks.

**Questions:**

1)	How would the method perform on omethods using different basis functions ([1,2]). Would the proposed limitations still exist?
2)	Since the paper argues that the limitations of Chebnet arises due to exploding response function with filter order: What if the norms would be constrained after every layer In Chebnet using layernorm, instancenorm etc. How would the method behave in this case? A study might help understand the importance of current method.
3) Please see weakness section above

[1] Xiyuan Wang, & Muhan Zhang. (2022). How Powerful are Spectral Graph Neural Networks.

[2] Mingguo He, Zhewei Wei, Zengfeng Huang, & Hongteng Xu. (2022). BernNet: Learning Arbitrary Graph Spectral Filters via Bernstein Approximation.

**Ethical Concerns:**

["NO or VERY MINOR ethics concerns only"]

**Final Justification:**

The author rebuttal has clarified most of the queries I had. Considering the response and other reviews, I keep my score of accepting the paper.

**Limitations:**

yes

**Quality:**

3

**Strengths And Weaknesses:**

**Strengths**

1)	The paper provides a novel analysis on ChebNet’s limitations and proposes a solution
2)	The method obtains competitive results over chebnet and current SoTA methods
3)	The paper is well written

**Weaknesses**

1)	While the overall method achieves good results, some studies should be provided on the benefit of the individual components. For example, the authors should provide ablations of their method using only antisymmetric weight, only the forward Euler method etc.
2)	The paper should also provide a study showing ability of the modified chebnet to approximate an arbitrary filter function. It would be interesting to see if it is able to better approximate the filters overcoming limitations of Chebnet here or whether some new limitations are introduced.

---

> ### Author Rebuttal · Authors · 2025-07-31
>
> We sincerely appreciate the reviewer’s insightful feedback on our work. Thank you for highlighting our novel analysis of ChebNet’s limitations and our proposed solution, as well as recognizing that our method delivers performance on par with other state‑of‑the‑art approaches. We’re also grateful for your kind words about the clarity and quality of our writing. Your comments and score motivate us to further strengthen and refine this research. Below, we address the reviewer’s comments in detail.
>
> **Extension to different basis functions:** Thank you for the insightful question. In this work, we focused on ChebNet due to its historical significance as the most well-known and widely used spectral GNN model. That said, a formal extension of our theoretical analysis to alternative bases (such as the ones discussed in [1,2]) would require revisiting the spectral norm and Jacobian dynamics in those settings. While we expect similar dynamics to emerge, a comprehensive treatment is outside the scope of this paper and is an exciting direction for future work. In theory, the non-dissipative aspect should generalize to other basis though we leave the analysis for separate work. Only then it would be interesting to address heterophily and compare different basis' ability to tackle a long-range, heterophilic graph given their ability to both propagate information in a stable manner and tackle different frequency levels depending on their filter properties which would prompt an interesting comparison across spectral basis (e.g Bernstein vs ChebNet).
>
> **Model behavior under normalization:** We appreciate the reviewer’s suggestion and agree that applying normalization techniques such as LayerNorm or InstanceNorm could help alleviate some of the instabilities observed in high-order ChebNet by implicitly bounding feature magnitudes. However, we would like to emphasize that the internal mechanisms of such normalization layers are complex, often dataset-dependent, and their stabilizing effect on model dynamics is not fully understood from a theoretical standpoint. In contrast, our proposed approach, based on Euler discretization and antisymmetric parameterization, is a principled and interpretable framework for enforcing stability, with formal guarantees on the Jacobian spectrum and resulting signal propagation properties of the model. That said, we agree that further comparison would be valuable. We will include a brief discussion of this point in the revised version of the paper, and we plan to investigate the empirical effect of normalization-based strategies as part of our camera-ready submission or future work.
>
> **Isolation antisymmetric effect:** Thank you for the insightful suggestion regarding component ablations. Following your recommendation, we conducted an additional experiment on the OGB‑Proteins benchmark to isolate the effect of antisymmetric weights in our model. Specifically, we evaluated a variant that enforces antisymmetry but omits the forward‑Euler discretization. This “antisymmetry‑only” model resulted in a significantly lower performance (72.2% test accuracy) compared to the full Stable‑ChebNet (79.5%), highlighting the importance of combining both antisymmetry and the discretization scheme for effective learning.
>
> | Model Variant           | Test Accuracy on OGB-Proteins |
> |-------------------------|-------------------------------|
> | Full Stable-ChebNet     | 79.5%                         |
> | ChebNet-Antisymmetry Only       | 72.2%                         |
>
> **On Stable-ChebNet’s ability function approximation ability:** We appreciate the reviewer addressing this concern. However, this point is not directly applicable to our method, as Stable-ChebNet does not alter the basis functions or the parameterization of the spectral filters used by ChebNet. The modification we propose applies solely to the propagation dynamics (i.e., how information flows through layers), by introducing a stable ODE-inspired formulation. As such, the filter expressivity and approximation capacity remain unchanged. We will make this clarification more explicit in the revised manuscript to avoid confusion on this point.
>
> We sincerely hope that our responses and revisions have addressed your concerns. We once again thank the reviewer for their valuable time and consideration.

---

> > ### Comment · Reviewer_zDKz · 2025-08-07
> > **Thanks for the rebuttal!**
> >
> > The author rebuttal has clarified most of the queries I had. I would request the authors to clarify these points in the main paper. Also, regarding the layer normalization, even if the results match the proposed method it would be worthwhile to study and the proposed method has merits despite the results.
> >
> > Best!

---

> > > ### Author Response · Authors · 2025-08-09
> > >
> > > We thank the reviewer for their engagement in the rebuttal process and for their valuable feedback.

---

### Official Review · Reviewer_E5Qi · 2025-07-02

**Clarity:** 4
**Significance:** 3
**Originality:** 2
**Rating:** 5
**Confidence:** 4

**Summary:**

The paper reevaluates the spectral GNN ChebNet on long-range graph tasks showing that it still performs competitively compared to baseline methods. It also introduces Stable-ChebNet, a slight modification of ChebNet to enforce layer-wise stability that achieves near state-of-the-art performance across a variety of experiments.

**Questions:**

1. Could you please elaborate why Figure 1 illustrates how classical ChebNet filters can exhibit unbounded dynamics? What do you mean by temperature here? Why does a temperature close to zero for most of the nodes at step T illustrate unbounded dynamics?
2. Overall Stable-ChebNet indeed achieves superior results on the synthetic long-range interaction datasets, but how can these findings translate to actual real-world datasets? The results from the long-range benchmark peptides dataset show that it can match the performance there, but the advantage over e.g. simple GCN is small. Do these results not suggest that we should just use more powerful methods like S2GCN, combining the two paradigms and be done with it?
3. Why did you evaluate ChebNet only on these three real-world datasets and did not include comparisons on standard homophilic node classification, the heterophilic datasets from Platanov et al. [3] and for the ones from OGB also on ogbn-products and some graph property prediction tasks? This would significantly simplify the inclusion of ChebNet as a standard baseline in future work.
4. The Barbell graph describes an extremely artificial scenario, which limits the transferability of these findings to real-world graphs. Could you elaborate further on how such a graph could arise in real-world scenarios?
5. Did you consider the findings and discussions from [1]? There has recently been a lot of discussion about fair and comparable evaluation of GNNs. They show that in many papers the baselines are not properly tuned. Do you think there is a similar issue with ChebNet, and did you use a similar approach? Or if not, could you use the ideas from [1] to improve the results of ChebNet even further (e.g. adding layer/batch norm)?

[1] Luo et al., Classic GNNs are Strong Baselines: Reassessing GNNs for Node Classification, NeurIPS 2024 Track on Datasets and Benchmarks

[2] Bamberger et al., Bundle Neural Network for message diffusion on graphs, ICLR 2025

[3] Platonov et al. A critical look at the evaluation of GNNs under heterophily: Are we really making progress?, ICLR 2023

**Ethical Concerns:**

["NO or VERY MINOR ethics concerns only"]

**Final Justification:**

Overall, the authors have addressed the concerns raised well, and the announced changes will further enhance the quality of the paper.

The main weakness of the paper lay in its empirical evaluation, which was initially insufficient for a practitioner to gain a clear understanding of the model’s overall performance. In the rebuttal, the authors provided additional results on both real and synthetic datasets, which substantially strengthened this aspect. In particular, for long-range datasets, I fully agree with the authors’ claim that (Stable-)ChebNet demonstrates a consistent advantage across a broad spectrum of tasks, underscoring its general practical relevance. Moreover, the results on other benchmark datasets provide clear indications that (Stable-)ChebNet can perform competitively in a wider range of scenarios.

Although certain points of discussion remain between the reviewer and the authors, particularly regarding benchmarking and the identification and assessment of long-range interactions, the overall benefits for the community are clear. Taking into account the discussion and the announced improvements, I therefore recommend accepting the paper.

**Limitations:**

yes

**Quality:**

3

**Strengths And Weaknesses:**

**Strengths**
- Paper is well-written, well-structured and most visualizations are helpful and clear.
- There are both theoretical and empirical validation for the claims made.
- The authors have successfully identified and filled a niche in the GNN literature/benchmarking. ChebNet is indeed often omitted as a baseline in the literature.
- It is clearly communicated why ChebNet should be included as a baseline in the future, especially on longe-range interaction graphs.

**Weaknesses**
1. Experiments are somewhat limited. The full results on LRGB and OGB could be provided.
2. Generally, the choice of methods to compare against is inconsistent. For example, in Table 3 and 4, completely different baselines are considered. Additionally, on long-range interaction tasks, one should also compare against methods like S2GCN or BuNN [2] that were created exactly for this task. Also, more recent experimental settings on standard GNNs like GAT from [1] are omitted, even though these would perform best on ogbn-proteins in the table provided.
3. The discussion on how to proceed based on these findings could be more pronounced. Yes, SpectralGNNs are strong baselines, but still if you look at the leaderboards of the benchmarks, e.g. on the OGB datasets, you don’t find any pure GNN approach anymore. How could this trend be changed?
4. Figure 5 and Table 2 seem to show the exact same information. Why were there no other baselines included that were constructed to work well for long-range interactions?

---

> ### Author Rebuttal · Authors · 2025-07-31
>
> We sincerely thank the reviewer for their encouraging feedback. We appreciate the recognition of the clarity and utility of our visualizations. We are also grateful for the acknowledgement of our theoretical and empirical contributions, as well as our effort to address a gap in GNNs.  Below, we address the reviewer’s comments in detail.
>
> **Clarifying the model’s figure:**  We apologize for the confusion around Figure 1;  To clarify, in this figure we plot at each layer the temperature of each node—by which we mean the ‘heat’ or magnitude of its signal—propagated through the filter. In the classical ChebNet (top row), high‑order Chebyshev filters act like a dissipative diffusion: by step T, most nodes have “cooled” almost to zero (very low temperature), indicating that information has effectively leaked away and that small perturbations can blow up or vanish uncontrollably—i.e. unbounded dynamics. In contrast, our Stable ChebNet (bottom row) enforces an antisymmetric, non‑dissipative update so that heat (information) is retained even at large T, yielding bounded, well‑conditioned propagation across layers. The main aim of this figure was to shed light on the importance of developing an information-preserving system by drawing an analogy to heat dissipation. We agree that “temperature” was not defined clearly in the original caption, and we’ll incorporate exactly this explanation in the corrected version.
>
> **Comparison with S2GCN and real world setting:** Although DRew and S2GCN achieve high AP on Peptides‑func, they both depend on expensive graph preprocessing steps: DRew must perform costly graph rewiring (e.g. all‑pairs shortest paths plus positional encoding construction) before training, while S2GCN requires a full Laplacian eigendecomposition for its spectral filter basis. These operations impose prohibitive runtime and memory overhead on large or evolving graphs.
> By contrast, Stable‑ChebNet delivers competitive long‑range accuracy entirely within the standard Chebyshev‑polynomials which approximate the Laplacian eigendecomposition: each layer executes sparse, polynomial‑filter updates in O(K·|E|) time, without any rewiring, eigencomputations, or additional encodings. This keeps training times and memory costs on par with a vanilla ChebNet, preserves end‑to‑end differentiability, and requires no extra implementation complexity, making Stable‑ChebNet both practically scalable and straightforward to deploy. The cost of using ChebNet compared to DRew is highlighted in Figure 2, which further motivates the need for a performant ChebNet on long-range tasks given its simplicity.
>
> While the community still lacks gold‑standard long‑range GNN benchmarks, we evaluated Stable‑ChebNet in complementary scenarios covering both graph and node-level tasks to demonstrate its applicability on real‑world and synthetic data. On the Peptides‑func benchmark, which highlights distant residue interactions, Stable‑ChebNet is on par with sota methods. On the OGB‑Proteins dataset where graphs can exceed 100k nodes, it achieves accuracy on par and often better than Graph Transformers, proving that stable information can compete with Transformers and other models of high complexity.
>
> **Testing on a newly released benchmark:** We note that very recently, Liang et al. [4] publicly released City‑Networks, a large‑scale transductive benchmark designed explicitly to probe long‑range dependencies in GNNs, derived from real-world city road networks.
>
>  Aligned with our findings, the benchmarking results of that paper show (though without detailed analysis) that ChebNet performs well on long-range tasks; it is often the best baseline and has a consistently high influence score, even on Ogbn-arxiv which we also report in our paper. To further validate our model on real-world road networks, we ran our Stable-ChebNet on the Paris City-Networks data and compared it to vanilla ChebNet in the Table below.
>
> | # of Layers (L) | K (Hops) | ChebNet (%)      | Stable-ChebNet (%)  |
> |-----------------|----------|------------------|---------------------|
> | 2               | 10       | 39.96 ± 0.04     | 49.52 ± 0.12        |
> | 3               | 10       | 47.16 ± 0.08     | 51.22 ± 0.07        |
> | 3               | 12       | 47.39 ± 0.12     | 51.28 ± 0.28        |
>
> These results underscore Stable‑ChebNet’s ability to capture long‑range dependencies in both synthetic and large‑scale real‑world settings using parameter‑efficient operations.
>
> **Additional benchmarks:** In line with the reviewer’s suggestion, we will additionally include results for the Bundle Neural Network (BuNN) from Bamberger et al. [2] across the common benchmarks, allowing for a more comprehensive comparison. In addition, we would like to clarify that our submission already includes a set of Graph Property Prediction tasks, namely Diameter, SSSP, and Eccentricity, designed to test long‑range information propagation (see Section 4 and Table 1). We believe these tasks are important to evaluate the long‑range capabilities of GNNs.
>
> **On the inclusion of heterophilic datasets:** We agree that examining the relationship between label heterophily and long‑range dependency is valuable.
> In this work, we do not benchmark on common heterophilic datasets as we believe there is not yet a clear connection between a graph being heterophilic and the presence of long-range dependencies — the latter being the main challenge this paper aims to address. One recent position paper sheds light on this issue [5], emphasizing that heterophily and long-range dependencies are not inherently linked as a graph can be heterophilic and require only local reasoning, or homophilic and require long-range reasoning.
>
> To further investigate this phenomenon, we have followed your recommendation and trained ChebNet and Stable‑ChebNet on the Roman‑Empire and Amazon‑Ratings datasets using the hyperparameters from Platonov et al. [3]. We find that both vanilla ChebNet and Stable-ChebNet can indeed match the highest performances reported in the Table with updated MPNN performances. Interestingly, we find that on both Amazon-Ratings and Roman-Empire, we converge towards higher performances at relatively lower hop neighborhoods, indicating a more local aspect to these heterophilic graphs.
>
> **Dataset: Amazon-Ratings**
> | K | L | Stable-ChebNet |
> |---|---|---------------|
> | 4 | 4 | 53.80 ± 0.32  |
> | 3 | 4 | 54.17 ± 0.11  |
> | 3 | 5 | 54.24 ± 0.23  |
>
>
> **Dataset: Roman-Empire**
> |  K  |  L  |     ChebNet      |       Stable-ChebNet       |
> |:---:|:---:|:----------------:|:------------------:|
> |  3  |  7  |  91.26 ± 0.36    |  90.65 ± 1.08      |
> |  5  |  7  |  90.79 ± 1.19    |  89.77 ± 1.06      |
> |  6  |  7  |  84.75 ± 0.35    |  89.03 ± 2.19      |
>
> **Emphasizing the importance of the Barbell dataset:**  While Barbell‐like structures might seem like a purely synthetic scenario, it actually captures a very common challenge in real‑world networks that are not well represented in the GNN community’s long-range benchmarks; One suitable example is two densely interconnected communities joined by just one or a handful of bridging connections. For instance, in infrastructure graphs we might have two urban centers whose local roads are very dense yet connected to each other by just one main highway. In protein‐interaction networks, functional modules such as signaling pathways can be highly interconnected internally, yet communicate with other modules through very few cross‐module interactions.
> These cases all mirror the barbell topology: high intra‐module cohesion plus a narrow inter‐module bottleneck. This creates exactly the same narrow bottleneck that standard GNNs struggle to traverse without squashing information. Hence, despite its simplicity, the barbell graph provides a useful stress test for how well GNN architectures can propagate information across real‐world “bridge” edges without oversquashing or loss of signal fidelity by just adjusting clique size.
>
> **The Role of Hyperparameter Tuning:** Thank you for your thoughtful suggestion and for pointing us to [1]. We have carefully considered the findings and discussions presented in that work. In fact, we had already examined [1] and noted the improvements in performance achieved through more rigorous hyperparameter tuning and architectural modifications such as the inclusion of additional layers or normalization techniques.
> At the time of our experiments, we adhered to the benchmark settings commonly used in prior work to ensure a fair and consistent comparison. While we did observe that deeper GNN architectures can lead to performance gains, we consciously operated within a parameter budget to highlight the efficiency of our model. We followed a similar protocol for Peptides func where we were bound to 500k params, above which we could reach near 72 AP on Peptides-func.
>
> Indeed, the improvements reported in [1], especially for GCN and GAT on the OGB-Proteins dataset—come with significantly increased parameter counts (on the order of ~1M and ~3M, respectively). In contrast, our ChebNet-based model uses only ~80K parameters, yet still manages to outperform even the 1M-parameter GCN variant on the same dataset. This demonstrates the effectiveness of our approach under a constrained parameter budget. That said, we acknowledge that incorporating ideas from [1] could further improve our model’s performance and plan to explore these enhancements in future work.
>
> We sincerely hope that our revisions have effectively addressed your concerns and encouraged a reconsideration of your evaluation and score.
>
> [4] Liang et al. (2025). Towards Quantifying Long-Range Interactions in Graph ML. arXiv preprint https://arxiv.org/abs/2503.09008
>
> [5] Adrián Arnaiz‑Rodríguez and Federico Errica (2025). Oversmoothing, “Oversquashing”, Heterophily, Long‑Range, and more: Demystifying Common Beliefs in Graph Machine Learning. Position paper published on https://arxiv.org/abs/2505.15547

---

> > ### Comment · Reviewer_E5Qi · 2025-08-04
> >
> > I thank the authors for their rebuttal and very much appreciate the efforts to further improve the quality of the paper. However, many of the responses only partially addressed my concerns or did not really respond to them.
> >
> > In general, the reply lacks structure and precision, i.e. there is no clear ordering or numbering, which makes them difficult to follow and evaluate. Upon careful review, I find that several of my original points, including W1, W2, W3, W4, Q2, and Q3, remain only partially answered or are still unaddressed. For example:
> >
> > 1. **Discussion of the bigger picture**
> >
> > Q2 did not concern Drew or computational efficiency. It specifically asked about S2GCN, because it combines spatial and spectral approaches and could therefore be seen as a generalization of purely spectral methods. As I tried to emphasize, while the results for ChebNet on many tasks were indeed missing, the more important point is: what conclusions can be drawn from the new evaluations? Ultimately, the focus should be on identifying the best-performing model for a given task; or, if model efficiency is the main concern, the best model on the Pareto front of performance versus efficiency. As seen e.g. for OGB, none of the usual GNNs achieve that (W3).
> >
> > Regarding W5, it is of course always to be welcomed if a method requires much less parameters to achieve comparable performance to larger baseline models. If this is one of the main claims about ChebNet, the paper could emphasize it more. However, another equally important factor today is whether a method can be *scaled up*. As discussed, [1] demonstrated that some widely-used GNNs can be scaled to models with millions of parameters (which is still relatively small for today’s standards). Investigating whether spectral methods can achieve similar scalability might be interesting for future works.
> >
> > 2. **New city-network datasets and long-range interactions**
> >
> > While the results on the new city-network datasets are certainly interesting, they do not enable the easy use of ChebNet as a baseline. Instead, they primarily reinforce ChebNet’s already well-demonstrated strength in handling long-range interactions. To better assess ChebNet’s suitability as a general baseline, it would be more valuable to evaluate its performance across the full range of classical graph benchmark datasets in the literature, as previously mentioned in Q3.
> >
> > Moreover, this new dataset seems to follow a questionable trend in benchmark *design*. It is often noted that existing benchmarks lack strong long-range interaction properties, but that does not mean we should construct datasets specifically to exhibit properties we wish to study. Rather, robust benchmarks should reflect a diverse set of real-world problems where models can be meaningfully tested. The new dataset seems to artificially introduce the need for long-range interactions through the use of eccentricity-based labels, aligning more closely with the purpose of a synthetic dataset. Synthetic datasets have a valid and important role, as they allow for controlled analysis of specific model capabilities. However, real-world *benchmarks* should focus on solving real problems on graphs that arise *naturally* in the different disciplines.
> >
> > 3. **Results on heterophilic datasets**
> >
> > The results on the heterophilic datasets from Platonov et al. [3], on the other hand, look really promising. They suggest that there are more reasons to use spectral GNNs then their ability to handle long-range interactions. Therefore, the paper could emphasize more the general role of (stable) ChebNet as a strong baseline model using these results.

---

> > > ### Author Response · Authors · 2025-08-06
> > > **Response (1/3)**
> > >
> > > We sincerely thank the reviewer for their time, valuable feedback, and re-evaluating our work.
> > >
> > > We provide detailed responses to the reviewer’s comments below:
> > >
> > > **Replying to the discussion on the bigger picture: (Point 1)**
> > >
> > > > Q2 did not concern Drew or computational efficiency. It specifically asked about S2GCN, because it combines spatial and spectral approaches and could therefore be seen as a generalization of purely spectral methods.
> > >
> > > Thank you for clarifying this.  We now understand that Q2 was aimed at evaluating S2GCN as a potential generalization of spectral methods, and we apologize for not addressing this in our original response.
> > >
> > > Combining spatial and spectral paradigms, as done in S2GCN, is indeed a promising direction. However, a major limitation of S2GCN lies in its reliance on cubic-time eigendecomposition, which severely hampers its scalability. This issue is not unique to S2GCN but also affects other rewiring-based methods such as DRew, which is why we referenced them together.
> > >
> > >  In contrast, our ChebNet-based approach (both vanilla and Stable-ChebNet) computes all updates via sparse polynomial filters in O(K·|E|) time per layer (where K is the Chebyshev order), without any eigencomputations. This keeps both runtime and memory efficient making it practical for large or evolving graphs.
> > >
> > > > what conclusions can be drawn from the new evaluations?
> > >
> > > In terms of some of the conclusions that can be drawn, we emphasize that our main goal with this paper is to highlight that ChebNet offers strong long-range propagation capabilities comparable to (or even surpassing) these methods, while being significantly more efficient. Despite this, ChebNet is rarely used as a baseline.
> > >
> > > A central motivation of our paper is to draw attention to this overlooked method and encourage further exploration of this type of efficient spectral approaches, particularly in the context of long-range modeling, where they may offer both performance and efficiency advantages.
> > >
> > > These findings do not go against the principles of S2GCN. In fact, they support them. We simply argue that the "spectral" part of the model could be conducted in an efficient and performant manner through ChebNet or similar approaches. We will add relevant discussion on S2GCN in the final version.
> > >
> > > **On scalability and depth**
> > >
> > > We agree with the reviewer that model scaling has been shown to be critical to performance. We would like to emphasize that a critical preconditioner to model scaling is the ability to make the models *go deep*. This aligns with some of the benchmarking in Luo et al. [1], which achieved these high parameter counts through the depth of the model.
> > >
> > >  One of the critical findings of our paper is that while Vanilla ChebNet offers good performance, its ability to scale to large depth is severely limited by its unstable signal propagation dynamics, which is why we designed a stable variant. We believe that some of the techniques used in Luo et. al  might implicitly be doing something similar through the use of residual connections and normalization. We will highlight these similarities in the revised version of the paper.
> > >
> > > **Regarding City-Networks (CityNet) (Point 2)**
> > >
> > >  >To better assess ChebNet’s suitability as a general baseline, it would be more valuable to evaluate its performance across the full range of classical graph benchmark datasets in the literature, as previously mentioned in Q3.
> > >
> > >  We do agree that their work does not specifically target ChebNet and its usage as a baseline, and that a more enhanced benchmarking on heterophilic datasets would better motivate a more general use of both Stable and vanilla ChebNet as benchmarks by the community. Having said that, we are suggesting CityNet as an addition rather than a replacement of the remaining heterophilic benchmarks.
> > >
> > >   >The new dataset seems to artificially introduce the need for long-range interactions through the use of eccentricity-based labels, aligning more closely with the purpose of a synthetic dataset.
> > >
> > > Thank you for the insightful remark. We would like to provide the following clarification concerning the CityNetworks dataset:
> > >
> > > - The road network does naturally arise from urban planning and is arguably more realistic than synthetically constructed graphs .
> > >
> > > - The task in this dataset is to predict the accessibility of locations in the road network i.e how much distance one needs to travel from one road junction to its 16-hop neighbours in the road network and goes beyond simply predicting eccentricity using graph topology alone.

---

> > > > ### Author Response · Authors · 2025-08-06
> > > > **Response (2/3)**
> > > >
> > > > Below is a follow up on the responses above :
> > > >
> > > >  >It is often noted that existing benchmarks lack strong long-range interaction properties, but that does not mean we should construct datasets specifically to exhibit properties we wish to study.
> > > >
> > > > Real-world graphs typically involve a combination of multiple factors, which makes it challenging to isolate the specific reasons for a model's failure. In our humble opinion, CityNetwork as clarified above does constitute a meaningful benchmark on long-range dependencies with a real-world context and task. Although long-range dependencies are not the sole factor to consider in practical graph scenarios, we believe the results obtained across the diverse datasets in this paper, particularly after the rebuttal, are encouraging. Nevertheless, we agree with the reviewer that it is important to consider other general baselines, and we have addressed this point in more detail below.
> > > >
> > > > **Updated baselines:** Following your suggestion during the rebuttal, we have expanded our evaluation to include additional benchmarks from Platonov et al. [3]. Specifically, while our original submission already included datasets such as **ogbn-arxiv** and **ogbn-proteins**, we have now incorporated **Amazon-Ratings** and **Roman-Empire** (see the tables in our previous response) to cover a broader homophily spectrum. Similarly, to further assess long-range information propagation, we have added **City-Networks (see table in our previous response) and Tree-Neighbors-Match** (see response to Rev EiU1) to complement the existing LRGB peptides, Barbell tasks, and Graph Property Prediction tasks.
> > > >
> > > > Therefore, our evaluation now comprises 12 different tasks including multiple homophilic levels and evaluation objectives, i.e., long-range propagation, homophilic and heterophilic node classification. Nevertheless, to ensure full coverage on LRGB, we will add PCQM in the next revision and report Stable-ChebNet’s performance there alongside Peptides. We believe these additions significantly strengthen the scope and generality of our experimental evaluation.
> > > >
> > > > **Point 3**
> > > >
> > > > Thank you for highlighting the strong performance on the Platonov heterophilic benchmarks, we will revise the manuscript to emphasize that Stable-ChebNet serves as a robust, general-purpose baseline across both homophilic and heterophilic settings. We’ll clearly position Stable-ChebNet not only as a long-range specialist but as a broadly competitive spectral GNN in diverse graph settings.
> > > >
> > > >
> > > > **Furthermore, to better address the weaknesses from the rebuttal, we have carefully reorganized our responses below. Each point is now clearly numbered and structured to directly correspond with your original comments. We hope this revised reply provides more clarity and completeness**
> > > >
> > > >
> > > > **Regarding Question 2**
> > > >
> > > > Across real-world benchmarks, the stability and propagation strengths from synthetic tasks consistently hold. On citation networks (ogbn-arxiv), Stable-ChebNet improves node classification by capturing long-range dependencies. For protein networks (ogbn-proteins), it matches or exceeds Graph Transformers like GraphGPS under tight parameter budgets.
> > > >
> > > > In e-commerce (Amazon-Ratings), it improves recommendation accuracy by capturing affinities missed by shallow methods. Finally, for road networks (CityNets), it preserves multi-hop distance estimates, turning synthetic robustness into real-world gains. In all cases, its stability mechanisms enable deeper, more reliable signal integration and consistent performance improvements.
> > > >
> > > >
> > > > **Regarding Question 3 and Weakness 1**
> > > >
> > > > We thank the reviewer for their suggestion. Following the rebuttal, we have added more benchmarks for a more comprehensive comparison (Our answer to Point 2 details the used dataset. The corresponding results are tabularized in the rebuttal).
> > > >
> > > > **Regarding Weakness 2- Tables 3 and 4**
> > > >
> > > > With regards to Tables 3 and 4, we consider the results from the available benchmarks on the corresponding datasets, and we try to include as many similar architectures such as Nodeformer and Spexphormer which are present in both, in addition to comparisons to a standard GCN. We are committed to making tables more homogeneous in a revised version.
> > > >
> > > > **Regarding Weakness 3- Changing trend on OGB**
> > > >
> > > > We share the reviewer’s observation that today’s leaderboards, especially on OGB, are dominated by hybrid architectures rather than spectral GNNs. We believe several advances can help shift this balance:
> > > >
> > > > - Deeper, well-conditioned propagation: Stable-ChebNet shows that with the right numerical safeguards, polynomial filters can be driven to high order without instability. Enabling 20–50 hop propagation in a pure GNN, while avoiding over-smoothing.
> > > >
> > > > - Efficient kernel approximations: Leveraging randomized or Lanczos‐based approximations boosts  scalability, allowing pure spectral GNNs to handle graphs with millions of nodes in a few seconds per epoch.

---

> > > > > ### Author Response · Authors · 2025-08-06
> > > > > **Response (3/3)**
> > > > >
> > > > > **Regarding Weakness 4**
> > > > >
> > > > >
> > > > > The reviewer is correct that, at first glance, Figure 5 and Table 2 are reporting the same barbell-graph MSE results for ChebNet vs. Stable-ChebNet. The rationale for including both is twofold:
> > > > >
> > > > > - Figure 5 gives a  qualitative, visual sense of how error grows with barbel size (N),showing the “partial collapse” regime versus the Stable-ChebNet’s robust performance across N (10, 25, 50).
> > > > >
> > > > > - Table 2 then provides the precise MSE values for key N (50, 70, 100) and filter orders K, so readers can see exact numbers (e.g. ChebNet’s MSE rises to ≈ 1.08 at N = 70 vs. Stable-ChebNet’s 0.06). These differences become less clear if shown in a plot like Figure 5.
> > > > >
> > > > > We compare against MPNN baselines from the BuNN paper for N=10, excluding Neural Sheaf Diffusion as it’s less relevant to our GNN-based method. However, we include GPS results from the updated BuNN paper. Unlike BuNN, which only reports N=10 (Table 2), we benchmark Barbell graphs from N=10 to 100. Notably, Graph GPS shows over-squashing even at N=10, unlike our model.
> > > > >
> > > > > We thank the reviewer once again for the insightful exchange. We genuinely hope that our responses and revisions have resolved your concerns and led you to reconsider your evaluation and score.

---

> > > > > > ### Comment · Reviewer_E5Qi · 2025-08-07
> > > > > >
> > > > > > Thank you for further engaging in the discussion and providing the detailed responses. I appreciate the clarifications, which I believe will help to further improve the quality of the paper. Many of my questions have been answered. Below, I would like to make some additional comments on specific points:
> > > > > >
> > > > > > **Benchmarking and datasets**
> > > > > >
> > > > > > I appreciate the inclusion of the PCQM dataset.
> > > > > >
> > > > > > I agree with the authors that twelve tasks were tested, around half of which are synthetic datasets. However, the paper primarily focuses on reevaluating an existing method (and a new variant of it). Therefore, extensive evaluation on real-world dataset is not only expected but also constitutes one of the paper’s main contributions. The existing evaluation of synthetic datasets showcases the long-range capabilities of ChebNet well, but as the authors stated themselves “Real-world graphs typically involve a combination of multiple factors, which makes it challenging to isolate the specific reasons for a model's failure.” This complexity is precisely why it is essential to test and compare methods on a variety of real-world datasets. For instance, while the synthetic results are informative, they do not explain why ChebNet achieves one of the best performances on the Roman-Empire dataset. Yet this is a valuable insight for the community.
> > > > > >
> > > > > > **Regarding City Networks**
> > > > > >
> > > > > > I would like to cite the paper [4] here:
> > > > > >
> > > > > > > Node labels based on eccentricity. To create class labels that require long-range dependency, we compute an approximation of eccentricity $\epsilon(v)$ in network science (Newman, 2018), which measures the maximum distance from a node $v$ to all other nodes in the graph […]. […] Instead of focusing on the entire graph, for each node $v$, we only consider its 16-hop neighborhood […]
> > > > > > >
> > > > > >
> > > > > > While I agree with the authors that the graph itself arises naturally, the clear intention of creating the eccentricity labels of the dataset was to require long-range dependencies. As I pointed out earlier, this is not inherently problematic, but it aligns more closely with the characteristics of a synthetic dataset.
> > > > > >
> > > > > > **Q2**
> > > > > >
> > > > > > > On citation networks (ogbn-arxiv), Stable-ChebNet improves node classification by capturing long-range dependencies.
> > > > > > >
> > > > > >
> > > > > > Is there any verification to support this statement? Specifically, how can we be certain that the observed performance improvements are indeed attributable to the model’s ability to capture long-range dependencies?
> > > > > >
> > > > > > **W2** I welcome the authors commitment to make the tables more homogeneous in a revised version.
> > > > > >
> > > > > > The inconsistency in the selection of baseline methods remains one of my main concerns with the paper, as it limits the comparability of results across datasets and complicates the use of the proposed method as a reliable baseline. For instance, Table 1 includes only GNNs and omits transformer models, despite one of the stated contributions being that ChebNet performs comparably to, or even better than, transformers while being more efficient. Additionally, given that these datasets are designed to assess a model's ability to capture long-range graph properties, it is unclear why established long-range methods such as S2GCN or BuNN are not included. Similar concerns apply to Figure 5 and Tables 3 and 4, as previously discussed.
> > > > > >
> > > > > > **W4** Thanks for the clarification regarding Figure 5 and Table 2.
> > > > > >
> > > > > > > However, we include GPS results from the updated BuNN paper.
> > > > > > >
> > > > > >
> > > > > > Where can I find this result?
> > > > > >
> > > > > > > Notably, Graph GPS shows over-squashing even at N=10, unlike our model.
> > > > > > >
> > > > > >
> > > > > > What does it mean for a transformer to show oversquashing? Since transformers generally work with global attention instead of local message passing, oversquasing is not well-defined in this context.
> > > > > >
> > > > > > What kind of analysis led to this result? Is this a shortcoming specific to the architecture of GPS, because of its hybrid structure, if yes what about other transformer architectures?

---

> > > > > > > ### Comment · Reviewer_EiU1 · 2025-08-07
> > > > > > >
> > > > > > > I thank both the Authors and Reviewer E5Qi for their thoughtful discussion. I believe that Reviewer E5Qi is raising very good points, and that the Authors are generally addressing them well.
> > > > > > >
> > > > > > > I would like to add a bit of nuance to the discussion on over-squashing in Transformers:
> > > > > > >
> > > > > > > > What does it mean for a transformer to show oversquashing? Since transformers generally work with global attention instead of local message passing, oversquasing is not well-defined in this context.
> > > > > > >
> > > > > > > It has recently been observed that over-squashing or representation collapse **can** be an issue in Transformers. Specifically, Barbero et al. [1] show that ``Transformer language models can lose sensitivity to specific tokens at the input, which relates to the well-known phenomenon of over-squashing'' Moreover, the GPS architecture includes message-passing layers after the attention layers, which could potentially exacerbate this issue.
> > > > > > >
> > > > > > > That said, it remains unclear whether over-squashing is the primary reason GPS fails at depths greater than 10. GPS is a relatively simple GT architecture, using both standard attention layers (without the architectural improvements seen in modern LLMs) **and** message-passing layers. Thus, while over-squashing is a plausible cause, I would suggest that the authors avoid attributing the failure solely to over-squashing, as multiple factors could be involved.
> > > > > > >
> > > > > > > Once again, I thank Reviewer E5Qi for their insightful comments. I strongly agree with their concerns regarding benchmarking practices and the inconsistency in baseline selection.
> > > > > > >
> > > > > > > [1]: Barbero et al., "Transformers need glasses! Information over-squashing in language tasks", NeurIPS 2024

---

> > > > > > > ### Author Response · Authors · 2025-08-07
> > > > > > >
> > > > > > > We thank the reviewer for their continued engagement and constructive feedback. We believe the discussion has contributed to improving the quality of our work. Below, we provide our responses to the follow-up comments.
> > > > > > >
> > > > > > > **Benchmarking and datasets**
> > > > > > >
> > > > > > > We are happy that the reviewer appreciated the inclusion of additional evaluation. We agree with the reviewer that evaluating models on real-world datasets is crucial to understand their practical effectiveness. We appreciate the reviewer’s point about Roman-Empire vs synthetic datasets, where high performance might not be trivially explained by long-range capabilities alone. We agree that further analysis in this direction would be valuable for the community and is a promising direction for future work.
> > > > > > > That being said, we believe our results currently highlight a consistent advantage of Stable-ChebNet across a broad spectrum of tasks, real and synthetic, supporting one of the leading points of the paper, that is its general practical relevance.
> > > > > > >
> > > > > > > **Regarding City Networks**
> > > > > > >
> > > > > > > We thank the reviewer for raising this point on the CityNet dataset.
> > > > > > >
> > > > > > > **Q2**
> > > > > > >
> > > > > > > Thank you for the question. We now agree with the reviewer that the exploitation of long-range dependencies alone is not sufficient for achieving strong performance on real-world tasks. However, we believe our claim is supported by the empirical evidence that, on ogbn-arxiv, Stable-ChebNet achieves approximately 2% higher accuracy compared to the standard ChebNet and it is 4% better than GCN. Given that the key architectural difference lies in the more stable and well-conditioned propagation mechanism, we attribute this improvement to a more effective retention and transmission of information, particularly over long distances in the graph. We also note that ogbn-arxiv has a graph diameter of 23, which suggests that long-range dependencies are potentially important.
> > > > > > >
> > > > > > > **W2**
> > > > > > >
> > > > > > > We’re glad the reviewer appreciates our effort to make the tables more homogeneous. Our initial choice of baselines was guided by the top-performing models on each benchmark leaderboard available at the time of our first submission. That said, we fully agree that having consistent baselines across experiments is important. We are currently running additional experiments to provide the most comprehensive comparison possible.
> > > > > > >
> > > > > > >
> > > > > > > **W4**
> > > > > > >
> > > > > > > > Where can I find this result?
> > > > > > >
> > > > > > > For your convenience, we report the results for GPS and BuNN for the Barbell task with N=10 below (originally found in Table 2 of the BuNN paper). We are currently running experiments to include results for all other node counts considered in our study.
> > > > > > >
> > > > > > > | Model | MSE ↓ (N=10) |
> > > > > > > |---|---|
> > > > > > > | GPS | 1.06 ± 0.13 |
> > > > > > > | BuNN | 0.01 ± 0.01 |
> > > > > > > | Stable ChebNet | 0.01 ± 0.00 |
> > > > > > >
> > > > > > > > What does it mean for a transformer to show oversquashing? Since transformers generally work with global attention instead of local message passing, over-squasing is not well-defined in this context. What kind of analysis led to this result? Is this a shortcoming specific to the architecture of GPS, because of its hybrid structure, if yes what about other transformer architectures?
> > > > > > >
> > > > > > > While transformers operate over fully connected graphs and are thus immune to distance-related bottlenecks, they still compress information from all other nodes into a single vector per node. This leads to a form of oversquashing, consistent with the definition in Alon & Yahav [6], where too much information is forced into fixed-size representations. As reviewer EiU1 notes, Barbero et al. (2024) demonstrate that even decoder-style transformers with causal masking suffer from oversquashing. Although their work does not focus on graph transformers, we believe the intuition applies and may help explain the poor performance of GraphGPS on the task in question.
> > > > > > >
> > > > > > >
> > > > > > > [6]: Alon & Yahav, "On the Bottleneck of Graph Neural Networks and its Practical Implications", ICLR 2021.

---

> > > > > > > > ### Comment · Reviewer_E5Qi · 2025-08-08
> > > > > > > >
> > > > > > > > I would like to thank the authors once again for their detailed responses, and also reviewer EiU1 for adding some nuance to the discussion on transformers. Overall, the authors have addressed the concerns raised well, and the announced changes will further enhance the quality of the paper.
> > > > > > > >
> > > > > > > > Although certain points of discussion remain between the reviewer and the authors, particularly regarding benchmarking and the identification and assessment of long-range interactions, the overall benefits for the community are clear. Taking into account the discussion and the announced improvements, I therefore see no obstacle for publication and will raise the score accordingly.

---

> > > > > > > > > ### Author Response · Authors · 2025-08-09
> > > > > > > > >
> > > > > > > > > We thank the reviewer for their time and valuable feedback which helps improve our work.

---

### Decision · Program_Chairs · 2025-09-17

**Decision:**

Accept (spotlight)

**Comment:**

All reviewers argue to accept this paper, which revisits a spectral approach to graph neural network modelling which was widely considered to be superseded by message passing neural networks and graph transformers. The paper includes novel analysis which motivates a more stable variant of the ChebNet, allowing scaling to greater depth.

There was very extensive discussion during the rebuttal phase, which led to greatly expanded experimental validation on a broader set of benchmarks, establishing the utility of the approach. Multiple reviewers raised their scores. I think this paper will be quite interesting to the community and may lead to broader interest and future work on spectral approaches to GNNs.